# Extracellularly Released Molecules by the Multidrug-Resistant Fungal Pathogens Belonging to the *Scedosporium* Genus: An Overview Focused on Their Ecological Significance and Pathogenic Relevance

**DOI:** 10.3390/jof8111172

**Published:** 2022-11-07

**Authors:** Thaís P. Mello, Iuri C. Barcellos, Ana Carolina Aor, Marta H. Branquinha, André L. S. Santos

**Affiliations:** 1Laboratório de Estudos Avançados de Microrganismos Emergentes e Resistentes (LEAMER), Departamento de Microbiologia Geral, Instituto de Microbiologia Paulo de Góes (IMPG), Centro de Ciências da Saúde (CCS), Universidade Federal do Rio de Janeiro (UFRJ), Rio de Janeiro 21941-901, Brazil; 2Rede Micologia RJ—Fundação de Amparo à Pesquisa do Estado do Rio de Janeiro (FAPERJ), Rio de Janeiro 21941-901, Brazil

**Keywords:** *Scedosporium*, *Lomentospora*, emergent fungi, extracellular molecules, enzymes, secondary metabolites

## Abstract

The multidrug-resistant species belonging to the *Scedosporium* genus are well recognized as saprophytic filamentous fungi found mainly in human impacted areas and that emerged as human pathogens in both immunocompetent and immunocompromised individuals. It is well recognized that some fungi are ubiquitous organisms that produce an enormous amount of extracellular molecules, including enzymes and secondary metabolites, as part of their basic physiology in order to satisfy their several biological processes. In this context, the molecules secreted by *Scedosporium* species are key weapons for successful colonization, nutrition and maintenance in both host and environmental sites. These biologically active released molecules have central relevance on fungal survival when colonizing ecological places contaminated with hydrocarbons, as well as during human infection, particularly contributing to the invasion/evasion of host cells and tissues, besides escaping from the cellular and humoral host immune responses. Based on these relevant premises, the present review compiled the published data reporting the main secreted molecules by *Scedosporium* species, which operate important physiopathological events associated with pathogenesis, diagnosis, antimicrobial activity and bioremediation of polluted environments.

## 1. Introduction: An Overview on the *Scedosporium* Genus

The *Scedosporium* genus is constituted of saprophytic filamentous fungi frequently isolated from human impacted environments, such as sewers, polluted waters, sediments, decaying vegetation, agricultural soils, hydrocarbon-contaminated soils, gardens, urban parks, playgrounds and hospital areas, compared to habitats with low human activity [1,2,3,4,5,6]. The *Scedosporium* genus is composed of the following species: *Scedosporium angustum*, *Scedosporium apiospermum*, *Scedosporium aurantiacum*, *Scedosporium boydii*, *Scedosporium cereisporum*, *Scedosporium dehoogii*, *Scedosporium desertorum*, *Scedosporium ellipsoideum*, *Scedosporium fusoideum* and *Scedosporium minutisporum* [7]. *Lomentospora prolificans*, formerly *Scedosporium prolificans*, was renamed due to its phylogenetic distance from the *Scedosporium* genus, as judged by both molecular and genetic parameters [8]. However, *L. prolificans* has been historically studied together with *Scedosporium* species; so, in this context, we decided to refer to scedosporiosis as the infection caused by both fungal genera in order to facilitate and to simplify the information.

*Scedosporium* species are emerging, opportunistic pathogens able to cause localized infections in immunocompetent individuals and disseminated infections in immunocompromised individuals [2]. Over the last few years, the number of cases of scedosporiosis has increased considerably, which may reflect, at least in part, an improvement in the diagnosis of its etiologic agents. For instance, the incidence of *Scedosporium* infection in a tertiary care cancer center in Texas (USA) per 100,000 patient–inpatient days increased from 0.82 cases between 1993 and 1998 to 1.33 cases from 1999 to 2005 [9]. Reviewing the literature, several publications have reported that the cases of disseminated scedosporiosis typically occurred in individuals undergoing hematopoietic stem cell transplantation (HSCT) and solid organ transplantation (SOT). In this context, the number of infections caused by these fungi accounted for approximately 25% of all non-*Aspergillus* mold infections in SOT recipients [10] and 29% of those in HSCT recipients, in which 75% of the infections in HSCT recipients and 61% of the infections in SOT recipients occurred within 6 months after transplantation [11]. The infection was disseminated in 69% and 46% of HSCT and SOT recipients with scedosporiosis, respectively [11]. Furthermore, the mortality rates in patients with disseminated *L. prolificans* infections are higher, up to 87.5% [12]. A study conducted by Heng and co-workers [13] revealed that scedosporiosis in hematology patients exerts a substantial impact on hospital resource consumption, length of stay and patient mortality, with the total costs (U$ 26,500.00 per patient) driven by ward stay and antifungal drug costs.

*Scedosporium* species also show a marked neurotropism and a high propensity to cause central nervous system (CNS) infections [2,14]. In human immunodeficiency virus (HIV)-positive patients colonized by *Scedosporium* spp., invasive scedosporiosis was proven in 54.5% of patients, with a mortality rate of 75%. In patients with CNS manifestations the mortality rate increases to 100% [15]. In an analysis of 99 cases of CNS infection caused by the *Scedosporium* genus, a similar percentage of mortality in immunocompetent and immunocompromised patients was reported (76% and 74%, respectively) [16]. Interestingly, CNS infection was preceded by near drowning or trauma in immunocompetent patients. Regarding the CNS infection in immunocompromised individuals, it was described as rapidly progressive disseminated lesions at various degrees of evolution [16]. Moreover, *Scedosporium* species rank second among the filamentous fungi most frequently isolated from cystic fibrosis patients, constituting a great risk factor for invasive infections for lung transplanted patients [17,18,19].

Infections caused by *Scedosporium* species are extremely difficult to treat because of the low susceptibility profile to all classes of antifungal drugs available for clinical use (e.g., azoles, echinocandins and polyenes). For *L. prolificans*, the scenario worsens since this species is pan-antifungal resistant [20]. At the moment, the treatment indicated for scedosporiosis is voriconazole together with surgical debridement when possible [21]. However, even when the recommendation is followed, the mortality rate is higher than 65% [20]. Thus, the relevance of *Scedosporium/Lomentospora* in the clinical scenario is obviously alarming due to both multiple antifungal-resistance and high morbimortality profiles.

Scedosporiosis usually starts with the inhalation or traumatic inoculation of conidial cells, which then germinate into hyphae that promote host cell/tissue invasion (Figure 1) [20,22]. The mycelial biomass formed by *Scedosporium/Lomentospora* species during the infection process resembles a typical biofilm structure, formed by a robust mass of hyphae surrounded by an extracellular polymeric matrix (Figure 2) [23,24,25,26,27,28,29]. The ability to form biofilms is essential for microbial cells to cope with environmental stress, host immunological responses and antimicrobial drugs [30]. The *Scedosporium* and *Lomentospora* biofilms are 2- to 1024-times more resistant to azoles (e.g., voriconazole), echinocandins (e.g., caspofungin) and polyenes (e.g., amphotericin B) than that observed in planktonic conidial cells. The increase in the resistance profile to antifungal drugs observed in biofilms is mainly due to the presence of the extracellular matrix, efflux pumps and the highly adapted response to oxidative stress [24,26,27,29]. The successful colonization by *Scedosporium* and *Lomentospora* species is partially due to the secretion of extracellular molecules that participate in nutrient acquisition, competition with other microorganisms, germination of conidia into hyphae and invasion of host cells and tissues, among other essential events [20,22].

The previously published reviews about *Scedosporium* species have addressed the pathogenesis mechanisms, immunology, treatment options, epidemiology, taxonomy and/or use of those species for bioremediation [6,20,22]; however, these works have never focused only on the different roles that extracellularly secreted molecules can play. In this context, herein we examine the available information about the extracellularly released molecules by *Scedosporium* species, including polysaccharides, non-peptide small-molecule metabolites, non-ribosomal peptides and (glyco)protein-nature molecules (Figure 3), as well as their potential roles in environmental colonization, successful host infection, nutrition, diagnosis and stress response. In addition, we reported for the first time on the effects of *Scedosporium* secretions on *Tenebrio molitor* larvae used herein as an in vivo model of infection.

## 2. Secretion: An Essential Biological Process in the Fungal Cell Cycle

The production of extracellular molecules is a universal process with fundamental importance in many aspects of the cellular physiology of all living cells, particularly fungi [31]. Throughout evolution, fungi have adapted their secretion machinery in order to perform a great number of specialized functions during different stages of the infectious process, allowing these microorganisms to cause illness [32,33]. The extracellularly released molecules play critical roles related to virulence and act at different stages of interaction, allowing fungal survival, multiplication and dissemination inside the infected host [33,34]. Moreover, it is known that secreted molecules can modulate the host immune response, helping the fungal cells to escape from the antimicrobial properties of antibodies, complementing proteins and antimicrobial peptides produced by the infected host [35]. For instance, the galactosaminogalactans secreted by mycelial cells of *Aspergillus fumigatus* are responsible for promoting fungal growth and survival inside the host due to their ability to trigger a Th2 immunosuppressive response [36]. Secreted/released molecules by fungi do not only interact with components and cells of the host immune system but can also induce different degrees of cytotoxicity on mammalian cells, being capable of activating distinct death pathways [37,38]. Some studies have shown that the exposure of host cells to fungal secretions is by itself sufficient to cause host cell death [39]. Schindler and Segal [40] showed that *Candida albicans* secreted metabolites directly affect the host cell cytoskeleton, inducing a rearrangement in actin filaments, which caused a decrease of 63% in the phagocytic activity of murine macrophages, resulting in the activation of apoptosis in these cells. Our research group showed that molecules secreted by mycelia of a clinical strain of *S. apiospermum* caused a significant loss of viability (CC_50_ = 0.24 μg/μL) in the confluent monolayer of pulmonary epithelial cells (A549 non-small-cell lung cancer cell line), inducing irreversible damage that begins with the rounding of the epithelial cells followed by their detachment from the plastic substrate (Figure 4) [35].

In a proteomic analysis performed before genomic sequencing of *S. apiospermum*, proteins involved in metabolic pathways (malate dehydrogenase, phosphomannomutase, triosephosphate isomerase, fructose-1,6-bisphosphate aldolase, phosphoglycerate mutase, mannitol-1-phosphate 5-dehydrogenase, aldose-1-epimerase, sterol metabolism-related protein), protein degradation/nutrition (aspartyl protease, haloacid dehalogenase-superfamily hydrolase), nucleotide metabolism (nucleoside diphosphate kinase), RNA processing (Ran-specific GTPase-activating protein 1), translation machinery (initiation factor 5a), morphogenesis (glucanase), transport (major facilitator superfamily multidrug transporter, Forkhead associated domain involved in signaling events and ABC-type transport system region), protection against stress (peroxiredoxin, heat shock protein, translationally controlled tumor protein, manganese superoxide dismutase), movement (cofilin, profilin and tropomyosin) and the allergen Asp f13-like protein were identified in *S. apiospermum* secretome [35]. In this regard, Figure 5 reveals the richness of polypeptides/proteins secreted by *Scedosporium*. Moreover, some of the secreted proteins were recognized by antibodies present in the serum obtained from a scedosporiosis patient, validating the role of secreted proteins during human infection [35]. The proteomic analysis of secreted molecules is currently being redone in our lab since the genome sequencing of *S. apiospermum* has become available [40,41]. It should be noted that the actual secretome analysis accounts for more than 120 distinct proteins (unpublished data), which drastically contrasts with the 25 previously identified proteins [35].

The larvae of the *Tenebrio molitor* beetle is currently being used as an in vivo model for studies on fungal infections, together with other invertebrate models such as *Drosophila melanogaster*, *Galleria mellonella* and *Caenorhabditis elegans*, due to the ease of manipulation and maintenance and low price, and as a valuable alternative to animal models [42]. In order to add more data about the effect of *Scedosporium* secreted molecules in the host, herein we present unpublished data about the cytotoxic effects of secreted proteins by three different strains of *S. apiospermum* in an in vivo model of *T. molitor* larvae; the experimental methodology is detailed in Figure 6. Proteins obtained from the strains after 7 days of mycelial culture were able to interfere with the viability of *T. molitor* in a typically dose-dependent manner (Figure 6B,C). These results demonstrate that *Scedosporium* secreted virulence factors are per se able to cause significant damage in the invertebrate host model.

**Figure 6 jof-08-01172-f006:**
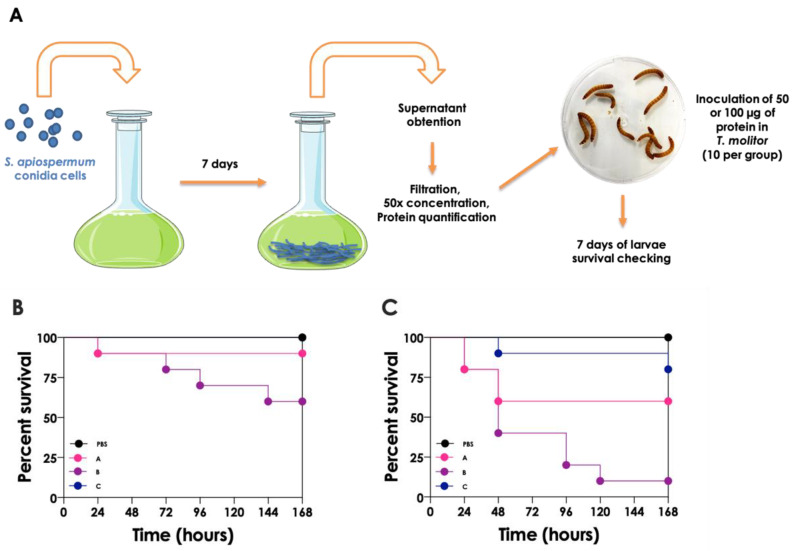
(**A**) Experimental scheme of *S. apiopermum* supernatant cytotoxicity test in *T. molitor*. Conidia of *S. apiospermum* (strains (**A**–**C**)) were inoculated in Sabouraud media and incubated for 7 days at 37 °C in constant agitation (120 rpm). Subsequently, supernatants obtained from 7-day cultures of different strains (**A**–**C**) were filtered in order to withdraw the remaining cells, concentrated 50 times utilizing a 10-kDa Amicon membrane, and the protein concentration was determined through the methodology developed by Lowry and co-workers [43]. Finally, *T. molitor* larvae (10 per group) were inoculated with cell-free supernatant containing either (**B**) 50 µg or (**C**) 100 µg of proteins, and the larvae survival was checked daily for 7 days. The survival data were plotted using the Kaplan-Meier method using GraphPad Prism 8.

## 3. Secretion of Specific Molecules by *Scedosporium*: Ecological, Physiological and Pathological Perspectives

### 3.1. Extracellular Vesicles: A Biological Carrier of Active Molecules

Despite the large number of studies on the detection and characterization of extracellular molecules in fungi, the secretion pathways in these microorganisms are particularly complex and not yet well understood, being largely even unknown [44]. In general, the secretion of molecules to the extracellular milieu by eukaryotic organisms can occur via conventional or unconventional/alternative pathways [45]. In the conventional pathway, proteins that will be secreted have an *N*-terminus-linked signal peptide to be incorporated into transport vesicles within the endoplasmic reticulum (ER) lumen and then directed to the cell surface through Golgi apparatus [46]. Proteins secreted in alternative routes lack the *N*-terminus-linked signal peptide and can reach the cell surface by multiple mechanisms, most of them by vesicles [46]. In fungal cells, molecules to be secreted must go through the entire thickness of the fungal cell wall, which provides additional complexity to the secretion process [47]. In recent years, the number of studies on the mechanisms developed by fungal cells to promote molecular transport through the cell wall has greatly increased. Nowadays, there are three non-exclusive hypotheses about how vesicles cross the fungal cell wall. First, enzymes could remodel the cell wall creating passages for vesicles through this physical barrier. Second, the existence of small pores in the cell wall suggests that these holes can also be used in vesicular transport. Another hypothesis would be mechanical pressure through cell wall pores [44,48,49]. Rodrigues and coworkers [50] described for the first time that fungi produce extracellular vesicles in vitro and in vivo, which are secreted through the cell wall, revealing that vesicular secretion is a key mechanism of extracellular delivery. So far, the extracellular vesicles have been described and characterized in several fungal species, such as: (i) yeasts of *Cryptococcus neoformans*, *C. albicans*, *Candida parapsilosis*, *Histoplasma capsulatum, Malassezia sympodialis, Paracoccidioides brasiliensis, Saccharomyces cerevisiae* and *Sporothrix schenckii*, (ii) protoplasts of *A. fumigatus*, (iii) conidia of *Aspergillus flavus*, (iv) hyphae of *Alternaria infectoria*, *Trichophyton interdigitale*, *Trichoderma reesei*, *Rhizopus delemar* and *Fusarium oxysporum* f. sp. *vasinfectum* [49,50,51,52,53,54,55,56,57,58,59,60]. The importance of producing vesicles by pathogenic fungi has been constantly proven. It is known that extracellular vesicles act as virulence pockets that deliver a concentrated load of fungal products directly to host cells and tissues (Rodrigues et al. 2008). For instance, extracellular vesicles produced by *C. neoformans* carry capsular components such as glucuronoxylomannan (GXM) and glucuronoxylomannangalactan (GXMGal), which are antigenic polysaccharides, as well as other virulence-associated components such as enzymes (e.g., urease, laccase and acid phosphatase), heat shock and antioxidant proteins, lipids and nucleic acids (DNA and RNAs) [61,62]. It has been reported that the vesicle components are recognized by serum antibodies from patients with cryptococcosis, histoplasmosis, paracoccidioidomycosis and candidiasis [51,53,61,63]. Moreover, the vesicular content is capable of inducing the production of several cytokines, such as tumor necrosis factor alpha (TNF-α), transforming growth factor beta (TGF-β) and interleukin-10 (IL-10), as well as activating nitric oxide (NO) production in murine macrophages in a typical dose-dependent manner [64].

In *S. apiospermum*, our research group observed the presence of extracellular vesicles through transmission electron microscopy images [35]. That report added *S. apiospermum* to the list of human pathogenic fungi capable of secreting extracellular vesicles, being the first filamentous fungal pathogen to join this list [35]. The secreted vesicles were detected leaving the fungal cells and entering the extracellular space in different parts of the cell wall, including the areas closest to the plasma membrane and in direct contact with the extracellular medium, in both conidial and mycelial forms of *S. apiospermum* (Figure 7). Moreover, the secretion of vesicles by *S. apiospermum* was also observed during the interaction of fungal particles with A549 lung epithelial cells (Figure 8) [65].

### 3.2. Hydrolytic Enzymes

Hydrolytic enzymes are fundamental in multiple processes of fungal life cycle and pathogenesis, including cell morphogenesis, nutrition, stress response, adhesion, invasion of cells/tissues and escape from immune system attack [66,67]. *Scedosporium* species are able to secret a wide range of enzymes, including proteolytic enzymes (e.g., serine, metallo, cysteine and aspartyl peptidases), lipases (e.g., esterase), DNAse, phosphatases (e.g., phytase) and many others (Figure 9). In this way, in this section we will explore the known hydrolytic enzymes secreted by *Scedosporium* species.

#### 3.2.1. Peptidases

Peptidases (or proteolytic enzymes) are degradative enzymes that catalyze the cleavage of peptide bonds in macromolecular proteins and oligomeric peptides. Peptidases are the single class of enzymes that occupy a pivotal position with respect to their applications in both physiological and commercial fields [68]. They are responsible for the complex processes involved in the normal physiology (e.g., nutrition, growth, differentiation) of the cell as well as in abnormal pathophysiological conditions (e.g., degradation of key host molecules like surface receptors and proteinaceous humoral response molecules, including antibodies, antimicrobial peptides and complement proteins) [66]. For instance, the secreted aspartic peptidases (Saps) produced by *C. albicans* and several non-*albicans Candida* species have essential roles in the hyphal formation and invasion of tissues through the degradation of collagen, fibronectin, laminin and mucin [32,69,70], whereas the secreted allergens Asp f5 (a matrix metallopeptidase) and Asp f13 (serine peptidase) by *A. fumigatus* are important for the recruitment of inflammatory cells and remodeling of the airways’ local immune response [71].

The first proteolytic enzyme described in *Scedosporium* spp. was a 33-kDa serine peptidase secreted by *S. apiospermum*, which presented optimum hydrolytic activity at pH 9.0 and temperatures between 37 to 50 °C, able to degrade human fibrinogen. The activity of this secreted serine peptidase was associated with the inflammation of the lungs of cystic fibrosis patients [72]. The production of distinct serine peptidases, belonging to the subtilisin-like, trypsin-like and elastase-like types, was also described in *S. aurantiacum* [73]. Interestingly, hypoxia conditions induced the secretion of these serine-type peptidases by *S. aurantiacum*; however, the main peptidases detected under these conditions were aspartic and cysteine peptidases, which presented optimum acidic pH for their full hydrolytic efficiency [74]. The aspartyl-type peptidase was also detected on the secretome of *S. apiospermum*, as corroborated with the use of either a specific substrate (cathepsin D) or peptidase inhibitor (pepstatin A) [35].

Metallopeptidases secreted by *S. apiospermum* presented differential profiles relying on the morphological type: mycelia were able to secrete six distinct peptidases ranging from 90 to 28 kDa, whereas conidia secreted a single peptidase of 28 kDa [75,76,77]. All metallopeptidases of *S. apiospermum* were active at acidic pH and completely inhibited by 1,10-phenanthroline, a classic inhibitor of metal-dependent enzymes [76,77]. These metallo-type peptidases are involved in the cleavage of key host proteins, such as immunoglobulin G, laminin, fibronectin and mucin [75,76,77], and they are also associated with the differentiation of conidial into hyphal forms [76,78]. Corroborating this last statement, 1,10-phenanthroline was able to completely block conidial germination, while ethylenediamine tetraacetic acid (EDTA) and ethylene glycol-bis(2-aminoethylether)-*N,N,N’,N’*-tetraacetic acid (EGTA), two well-known metal chelators, only partially inhibited the conidial germination [33,78].

Peptidases secreted by *Scedosporium* species were able to cause morphological changes in epithelial cells, detachment of the monolayer and reduce their viability [35,79]. In addition to planktonic cells, the production of hydrolytic enzymes has also been described in biofilm-forming cells of *Scedosporium* species. Biofilms formed for 72 h by *S. apiospermum, S. minutisporum, S. aurantiacum* and *L. prolificans* released peptidases able to hydrolyze albumin and casein at pH values ranging from 4.0 to 9.0 [80].

#### 3.2.2. Lipases

Lipases are a class of enzymes that catalyze the hydrolysis of triglycerides to glycerol and free fatty acids, as well as the hydrolysis and transesterification of other esters [81]. These properties make microbial lipases relevant in several industries (e.g., food, chemical and pharmaceutical) and also for the infectious process through the hydrolysis of host components (e.g., plasma membrane) [81,82].

Studies about the secretion of lipases by *Scedosporium* species have focused on the use of this hydrolytic enzymes in industrial and bioremediation applications (which are better detailed in sub item 3.5). For instance, *S. boydii* secretes extracellular lipases able to biodegrade a triacylglycerol named tributyrin, as well as linseed, olive and soybean biodiesel [83,84]. Some studies have demonstrated that *Scedosporium* species are able to penetrate the membrane of mammalian cell lines [85,86,87], which indicates the secretion of phospholipase as observed in *L. prolificans* isolates from Mexican patients [88]. In addition, 72 h-biofilms of *S. apiospermum, S. minutisporum, S. aurantiacum* and *L. prolificans* produced lipases able to hydrolyze different lipid chain sizes, such as 4-methylumbelliferylbutyrate, 4-methylumbeliferyl heptanoate and 4-methylumbeliferyl oleate [80].

#### 3.2.3. Phosphatases

Phosphorylation is an essential step for several processes in living cells, including cell cycle progression, central metabolism reactions, filamentation and gene transcription, among others [89]. Phosphorylation levels are coordinated by a fine balance between protein kinases and phosphatases in response to internal and external signals [89].

Acid and alkaline phosphatases were identified on the mycelial surface of *S. apiospermum* and also in the secretome of this species as demonstrated with specific inhibitors (levamisole, inorganic phosphate, sodium orthovanadate, ammonium molybdate, sodium fluoride, sodium β-glycerophosphate and sodium tartrate) and cytochemical localization [35,90]. The roles of these phosphatases in virulence are mostly unknown for *Scedosporium*, but for other fungal species these enzymes are involved in virulence traits, such as adhesion to epithelial cells and biofilm formation [35,90,91,92]. It is an open and promising area for future research.

#### 3.2.4. Detoxifying Enzymes

To colonize the host, pathogens have to face toxic oxygen and nitrogen species produced by phagocytic cells and other threats [93]. One way to cope with such stressors is the production of antioxidant enzymes, such as catalase and superoxide dismutase (SOD). SODs are metalloenzymes that are on the front line of defense against oxidative stress, catalyzing the conversion of superoxide anion (O2^•−^) to H_2_O_2_ and oxygen molecules [93,94]. In fungal cells, SODs are found mainly intracellularly on both cytosol and mitochondria; however, few SODs are extracellularly detected [93].

Studies about SOD in *Scedosporium* species are scarce. Initially, a cytosolic Cu,Zn-SOD was characterized in *S. apiospermum*. The production of this enzyme was stimulated by iron starvation, and it was not detected on the fungal culture filtrates with the methodology used [94]. Subsequently, a glycosylphosphatidylinositol-anchored SOD detected on the surface of *S. apiospermum* conidial cells was described; however, the secretion of this enzyme was not evaluated [93]. Notably, a Mn-SOD was identified in the secretome of *S. apiospermum*, indicating that at least one SOD can be secreted and probably has an extracellular function [35].

Catalases act by decomposing H_2_O_2_ into molecular oxygen and water [27]. In *Scedosporium*, the catalases A1, A2 and A2′ were identified in mycelial extract, and the presence of catalase A1 in the extracellular milieu was also observed [95,96]. Due to its antigenic nature, the authors of that study proposed the use of catalase A1 as a diagnostic tool for *Scedosporium* species, which will be further detailed herein.

Interestingly, a peroxiredoxin was also detected on the secretome of *S. apiospermum* [35]. The gene encoding a peroxiredoxin (*SaPrx2*) is overexpressed when *S. apiospermum* is in a co-culture with phagocytic cells (THP1 and HL60), as well as upon exposure to menadione and H_2_O_2_ [97].

#### 3.2.5. Mechanisms to Obtain Iron

Iron is an essential micronutrient for all organisms involved in several cellular processes; however, iron is not freely available because it is mainly under ferric state [98]. In this way, microorganisms have developed some mechanisms to obtain iron from the environment, including the secretion of hemolysins and siderophores.

Hemolysins are molecules that cause the lysis of red blood cells by disrupting the cell membrane, allowing iron acquisition from hemoglobin [99]. The hemolytic activity of *Scedosporium* species is most unknown, but herein we demonstrated the hemolysin activity of *S. apiospermum* through the presence of a halo around the fungal colony when grown on Sabouraud medium supplemented with 7% sheep blood (Figure 10). In addition, high hemolysis has been identified as a symptom of a patient with scedosporiosis [100].

Siderophores are small organic molecules able to scavenge iron in iron-restricted environments and/or environments with competition for this micronutrient [98]. *S. apiospermum* secreted two siderophores, dimerumic acid and *N*𝛼-methyl coprogen B, both from the hydroxamate type and coprogen family [98]. The production of the *N*𝛼-methyl coprogen B is higher for clinical isolates from respiratory samples compared to environmental strains, suggesting its use as a promising diagnostic tool, as will be detailed in Section 3.3 [98,101]. Moreover, the *N*𝛼-methyl coprogen B is essential for fungal growth and virulence, as demonstrated by its blockage synthesis due to the disruption of the *sidD* gene [102].

### 3.3. Molecules Related to Fungal Diagnosis

Traditional diagnosis methods, such as mycological examination, can be ineffective for *Scedosporium* species in some cases, such as in polymicrobial clinical samples, due to their slower growth compared to other filamentous fungi (e.g., *A. fumigatus*). Another relevant issue in this regard is the morphological similarity of *Scedosporium* species to other hyaline filamentous fungal species (e.g., *Aspergillus* spp. and *Fusarium* spp.) in histopathological tissue sections [2,96]. For successful patient treatment, the correct *Scedosporium* identification is essential, since amphotericin B is the first line of treatment for several infections caused by filamentous fungi and *Scedosporium* species are intrinsically resistant to this antifungal agent [103]. In this context, immunological diagnosis emerged as a prominent option; however, immunological *Scedosporium* diagnosis has focused for long time on the use of polyclonal antibodies in counterimmunoelectrophoresis and immunohistological techniques, for which positive results can be due to a cross-reaction with antigens from other clinically relevant fungal species [2,104,105]. In this manner, the diagnosis of *Scedosporium* species through the identification of specific extracellular and/or secreted molecules arises as a promising research field. The peptide-polysaccharide called peptidorhamnomannan (PRM) was the first molecule from *S. apiospermum* (formerly *Pseudallescheria boydii*) suggested to be used in diagnosis [106]. PRM is composed of Rhap(1 → 3)Rhap on side chains, which may be (1 → 3) to (1 → 6)-linked mannose units, and is found on the surface of both conidial and mycelial cells of *S. apiospermum, S. boydii, S. minutisporum, S. aurantiacum* and *L. prolificans*; moreover, this molecule was also detected on the extracellular milieu of *S. apiospermum* growth, indicating its secretion [65,85,106,107,108,109]. PRM strongly reacts with an antiserum obtained against the whole *S. apiospermum* cell utilizing enzyme-linked immunosorbent assay (ELISA) and immunofluorescence techniques (Figure 11), but it reacts poorly against an antiserum obtained with *Sporothrix schenckii* (which also produces a PRM-like molecule), demonstrating that PRM can be used in the differential diagnosis of *Scedosporium* spp. [106]. In addition to its use in diagnosis, PRM is an essential molecule used by the fungal cells to interact with mammalian cells, such as larynx epithelial carcinoma (HEp2) and macrophages, as well as for the induction of pro-inflammatory cytokine production. Moreover, the use of monoclonal antibodies (mAbs) against PRM on the surface of conidial cells decreases phagocytosis and the chemical removal of *O*-linked oligosaccharides from PRM abolished pro-inflammatory cytokines [85,107,108,110]. Notably, the extracted PRM from the surface of *L. prolificans*, *S. apiospermum*, *S. boydii* and *S. aurantiacum* is able to inhibit the growth and biofilm formation of *Staphylococcus aureus*, *Burkholderia cepacia* and *Escherichia coli* [108], revealing its antibacterial activity.

Subsequently, IgM and IgG1 K-light chain mAbs were developed. These mAbs recognize a carbohydrate epitope on an unknown extracellular 120-kDa antigen present on the *S. apiospermum* conidial and mycelial surface, and also present in fungal culture filtrates [103]. The mAbs specifically identified *S. apiospermum* and no other clinically relevant fungi, such as *A. fumigatus, C. albicans, C. neoformans, Fusarium solani* and *Rhizopus oryzae*, in immunofluorescence and double-antibody sandwich enzyme-linked-immunosorbent assays. However, these mAbs also react with *Graphium* and *Petriella* species [103].

In addition to diagnosis through the identification of cell surface components that can be secreted, another alternative approach to diagnosis is the detection of specific metabolites secreted by microorganisms during the infection course [101]. For instance, siderophores and pseudacyclins were suggested as good options for the identification of *Scedosporium* species on clinical samples [98,101,111]. As mentioned before, siderophores are secreted in environments with iron competition or scarce concentration, such as in cystic fibrosis (CF) sputum [98,101]. The siderophore *N*𝛼-methyl coprogen B was identified as specifically secreted by *Scedosporium* species among CF-related microorganisms and, consequently, a marker of *Scedosporium* colonization [101]. For this reason, its use has been suggested for *Scedosporium* diagnosis in CF patients. Utilizing high performance liquid chromatography, this siderophore was only identified in *S. apiospermum* supernatant, and not on *Aspergillus* spp. and *Exophiala dermatiditis*. In addition, the siderophore was only detected in sputum from CF patients colonized by *Scedosporium* spp. [101]. Likewise, five cyclic peptides with an unknown role in metabolism, named as pseudacyclins A-E, are produced exclusively by *Scedosporium* species and have been patented (International Publication Number: WO 2009/149675 A3) to be used as a diagnostic tool for the *Scedosporium* genus [111,112].

Enzymes are also molecules suggested for use in *Scedosporium* species diagnosis. As mentioned before, detoxification enzymes, such as catalase and SOD, are produced during all infection courses by microorganisms in order to face the reactive oxygen species produced by host phagocytic cells [96]. The catalase A1, a tetrameric protein of 460 kDa, is an enzyme present both in mycelium and in the culture supernatant, which is recognized by the serum of CF patients with scedosporiosis, but not with *A. fumigates,* through an ELISA assay [96,113].

The detection of 1,3-β-D-glucan is a traditional methodology in the diagnosis of invasive fungal infection [114]. This molecule is found in the fungal cell wall and is also secreted, which makes possible its detection in the blood of patients with invasive fungal infections [115]. The detection of 1,3-β-D-glucan has been reported in cases of brain abscess and invasive diseases caused by *Scedosporium/Lomentospora* species [115,116,117,118].

### 3.4. Secondary Metabolites

In fungi, secondary metabolites are derived from central metabolic pathways, being the acyl-CoAs the initial mole-cules for synthesis [119]. In contrast, non-ribosomal peptides (NRPs) are synthesized by NRP synthases initiating in amino acids [119,120]. An analysis of NPR synthases’ gene clusters in *S. apiospermum* identified nine putative NRPS clusters involved in the synthesis of epidithiodioxopiperazines, siderophores, cyclopeptides or other still uncharacterized specialized metabolites [121]. Secondary metabolites play essential roles in cell development and interaction with other organisms [119]. For example, some of the most relevant secondary metabolites are the β-lactam antibiotics, such as penicillins and cephalosporins, produced by fungi belonging to the *Penicillium, Cephalosporium* and *Aspergillus* genera, as well as by bacteria of the genera *Streptomyces, Nocardia, Flavobacterium* and *Lysobacter* [120,121].

Secondary metabolites secreted by *Scedosporium* species have many biological activities, such as antitumor, antimicrobial, insecticidal and antidiabetics (Table 1). The tyroscherin produced by *Scedosporium* spp. Presented an in vitro selective antitumor activity against insulin-like growth factors-dependent cell lines, such as the human breast cancer cell lines MCF-7 and T47D, with half maximal inhibitory concentration (IC_50_) of 9.7 ng/mL and 32 ng/mL, respectively [122]. The molecules ovacilin and boydone B produced by *S. boydii* present antitumor activity against the lung cancer cell line A549 with an IC_50_ of 4.1 and 41.3 µM, respectively [123]. The 3,3′-cyclohexylidenebis(1H-indole) is a secondary metabolite produced by *S. boydii*, which had its activity tested against a greater number of cancer cells: human lung cancer cell lines (A549 and GLC82), human nasopharyngeal carcinoma cell lines (CNE1, CNE2, HONE1 and SUNE1) and human hepatoma carcinoma cell lines (BEL7402 and SMMC7721), with IC_50_ values ranging from 18.69 to 27.53 µM [124]. Secondary metabolites produced by *Scedosporium* also have antidiabetic activity. The quinadoline A and the scequinadoline D, E and J promote triglyceride accumulation in 3T3-L1 preadipocytes cells, through induction of adipogenesis [125].

The worldwide emergence of fungal and bacterial resistant strains to antimicrobial drugs has highlighted the current low arsenal of antimicrobials to deal with some types of microbial infections [126,127]. For example, infections caused by resistant *Klebsiella pneumoniae* strains, as well as azole-resistant *Aspergillus* and *Candida* species, are associated with significant morbidity and mortality due to the lack of optimal treatment [126,128]. An estimate made in 204 countries showed that in 2019 the number of deaths associated with antimicrobial resistance was 4.95 million [129]. In this context, research into new antimicrobial compounds has intensified in recent decades, and secondary metabolites produced by fungal and bacterial cells are emerging as promising candidates, since microbial cells secrete bioactive compounds able to inhibit other microorganisms during the competition for nutrients [130]. Several secondary metabolites produced by *Scedosporium* species with antimicrobial activity have been identified; among them are compounds with antibacterial activity (Table 1; Figure 12). In this context, the *Scedosporium* metabolites gliotoxin, dehydroxybisdethiobis(methylthio)gliotoxin, *bis*dethio*bis*(methylthio)gliotoxin, fumitremorgin C, 12,13-dihydroxyfumitremorgin and boydone A had antibacterial activity against *S. aureus*, including methicillin-resistant strains [130,131,132]. The *Scedosporium* secondary metabolites also have antifungal and antiviral activities. The named “inhibitory compound” produced by *S. boydii* has a fungistatic activity against *Alternaria brassicicola*, reducing the disease incidence of black leaf spot of spoon cabbage [133]. Moreover, the secondary metabolites tyroscherin and *N*-methyltyroscherin were effective against *C. albicans, C. neoformans, A. fumigatus* and *Trichophyton rubrum* [134,135]. The secondary metabolites scequinadoline D and scedapin C displayed anti-hepatitis C virus activity with an effective concentration of 110.35 and 128.60 µM, respectively [136]. Moreover, some secondary metabolites (e.g., diketopiperazines pseudoboydone C, cyclo-(Phe-Phe), cyclopiamide E, 24,25-dehydro-10,11-dihydro-20-hydro-xyaflavinin and aflavinine) secreted by *S. boydii* had insecticidal activity against a major agricultural pest insect, *Spodoptera frugiperda* [137]. Other biological functionalities described for secondary metabolites secreted by *Scedosporium* spp are: inhibitor of acyl-CoA (e.g., AS-183), stimulator (e.g., pseurotin A) or inhibitor (e.g., (-)-ovalicin) of osteoclastogenesis; in addition to several other molecules that as yet have some undefined biological activity (Table 1; Figure 12) [123,124,137,138,139,140,141,142,143,144,145,146,147,148,149,150,151].

**Table 1 jof-08-01172-t001:** List of secondary metabolites identified in *Scedosporium* species and their potential biological activity.

Species	Molecule	Activity	References
*Scedosporium* spp.	AS-183	Inhibitor of acyl-CoA	[138]
*S. ellipsoidea*	YM-193221	Antifungal	[134]
*Scedosporium* spp.	Tyroscherin	Antitumor; Antifungal	[122,135,139]
*Scedosporium* spp.	Dehydroxy*bis*dethio*bis*(methylthio)gliotoxin	Antibacterial	[131]
*Scedosporium* spp.	*Bis*dethio*bis*(methylthio)gliotoxin	Antibacterial	[131]
*Scedosporium* spp.	Gliotoxin	Antibacterial	[131]
*Scedosporium* spp.	12,13-hihydroxyfumitremorgin C	Antibacterial	[132]
*Scedosporium* spp.	Fumitremorgin C	Antibacterial	[132]
*Scedosporium* spp.	Brevianamide F	Antibacterial	[132]
*Scedosporium* spp.	(2*RS,*8*R*,10*R*)-YM-193221	ND	[139]
*S. boydii*	“Inhibitory substance”	Antifungal	[133]
*S. boydii*	Pseudallin	Antibacterial	[152]
*S. boydii*	Boydone A	Antibacterial	[123,130]
*S. boydii*	Boydone B	Antitumor	[123]
*S. boydii*	Botryorhodine F	ND	[123]
*S. boydii*	Botryorhodine G	ND	[123]
*S. boydii*	Fusidilactone A	ND	[123]
*S. boydii*	(*R*)-(-)-mevalonolactone	ND	[123]
*S. boydii*	(*R*)-(-)-lactic acid	ND	[123]
*S. boydii*	Ovalicin	Antitumor	[123]
*S. boydii*	Botryorhodine	ND	[123]
*S. boydii*	*N*-methyltyroscherin	Antifungal	[135]
*S. boydii*	Boydine A	ND	[140]
*S. boydii*	Boydine B	Antibacterial	[137,140]
*S. boydii*	Boydine C	ND	[140]
*S. boydii*	Boydine D	ND	[140]
*S. boydii*	Boydene A	ND	[140]
*S. boydii*	Boydene B	ND	[140]
*S. boydii*	Pseudaboydin A	ND	[137]
*S. boydii*	Pseudaboydin B	ND	[137]
*S. boydii*	(R)-2-(2-hydroxypropan-2-yl)-2,3-dihydro-5-hydroxybenzofuran	ND	[137]
*S. boydii*	(R)-2-(2-hydroxypropan-2-yl)-2,3-dihydro-5-methoxybenzofuran	ND	[137]
*S. boydii*	3,3′-dihydroxy-5,5′-dimethyldiphenyl ether	ND	[137]
*S. boydii*	3-(3-methoxy-5-methylphenoxy)-5-methylphenol	ND	[137]
*S. boydii*	(-)-Regiolone	ND	[137]
*S. boydii*	6-Chloro-2-(2-hydroxypropan-2-yl)-2,3-dihydro-5-hydroxybenzofuran	ND	[141]
*S. boydii*	7-Chloro-2-(2-hydroxypropan-2-yl)-2,3-dihydro-5-hydroxybenzofuran	ND	[141]
*S. ellipsoidea*	Pseudellone A	ND	[142]
*S. ellipsoidea*	Pseudellone B	ND	[142]
*S. ellipsoidea*	Pseudellone C	ND	[142,145,146]
*S. ellipsoidea*	Pseudellone D	ND	[145]
*S. ellipsoidea*	(5S,6S)-dihydroxylasiodiplodin	ND	[145]
*S. ellipsoidea*	Lasiodipline F	ND	[145]
*S. ellipsoidea*	(5S)-hydroxylasiodiplodin	ND	[145]
*S. boydii*	Pseuboydone A	ND	[137]
*S. boydii*	Pseuboydone B	ND	[137]
*S. boydii*	Diketopiperazines pseudoboydone C	Insecticidal	[137]
*S. boydii*	Diketopiperazines pseudoboydone D	ND	[137]
*S. boydii*	Haematocin	ND	[137]
*S. boydii*	Phomazine B	ND	[137]
*S. boydii*	Bisdethiobis(methylthio)gliotoxin	ND	[137]
*S. boydii*	Cyclo-(2,20-dimethylthio-Phe-Phe)	ND	[137]
*S. boydii*	Cyclo-(Phe-Phe)	Insecticidal	[137]
*S. boydii*	Ditryptophenaline	ND	[137]
*S. boydii*	Speradine B	ND	[137]
*S. boydii*	Speradine C	ND	[137]
*S. boydii*	Cyclopiamide E	Insecticidal	[137]
*S. boydii*	24,25-Dehydro-10,11-dihydro-20-hydro- xyaflavinin	Insecticidal	[137]
*S. boydii*	Aflavinine	Insecticidal	[137]
*S. boydii*	b-Aflatrem	ND	[137]
*S. boydii*	Pyripyropene A	ND	[137]
*S. boydii*	Pseudoscherine	ND	[137]
*S. boydii*	4-(1-Hydroxy-1-methylpropyl)-2-isobutyl-pyrazin-2(1H)-one	ND	[137]
*S. boydii*	4-(1-Hydroxy-1-methyl-propyl)-2-secbutylpyrazin-2(1H)-one	ND	[137]
*S. boydii*	*O*-methyl sterigmatocystin	ND	[137]
*S. boydii*	Asperfuran	ND	[137]
*S. boydii*	Pseudallicin A	ND	[147]
*S. boydii*	Pseudallicin B	ND	[147]
*S. boydii*	Pseudallicin C	ND	[147]
*S. boydii*	Pseudallicin D	ND	[147]
*S. apiospermum*	Scedapin A	ND	[148]
*S. apiospermum*	Scedapin B	ND	[148]
*S. apiospermum*	Scedapin C	Antiviral	[148]
*S. apiospermum*	Scedapin D	ND	[148]
*S. apiospermum*	Scedapin E	ND	[148]
*S. apiospermum*	Scequinadoline A	ND	[148]
*S. apiospermum*	Scequinadoline B	ND	[148]
*S. apiospermum*	Scequinadoline C	ND	[148]
*S. apiospermum*	Scequinadoline D	Antiviral	[148]
*S. apiospermum*	Scequinadoline E	ND	[148]
*S. apiospermum*	Scequinadoline F	ND	[148]
*S. apiospermum*	Scequinadoline G	ND	[148]
*S. boydii*	Pseurotin A	Stimulatory osteoclastogenesis	[143]
*S. boydii*	(-)-Ovalicin	Inhibitory osteoclastogenesis	[143]
*S. boydii*	Chlovalicin	ND	[143]
*S. boydii*	Dihydroxybergamotene	ND	[143]
*S. boydii*	AM6898B	Stimulatory osteoclastogenesis	[143]
*S. boydii*	Aspergiketone	ND	[143]
*S. boydii*	Pseudboindole A	ND	[124]
*S. boydii*	Pseudboindole B	ND	[124]
*S. boydii*	3,3′-Cyclohexylidenebis(1H-indole)	Antitumor	[124]
*S. boydii*	Indole alkaloids	ND	[124]
*S. boydii*	2-Hydroxy-2-(propan-2-yl) cyclobutane-1,3-dione	ND	[149]
*S. apiospermum*	Scetryptoquivaline A	ND	[125]
*S. apiospermum*	Quinadoline A	Antidiabetic	[125]
*S. apiospermum*	Scequinadoline D	Antidiabetic	[125]
*S. apiospermum*	Scequinadoline E	Antidiabetic	[125]
*S. apiospermum*	Scequinadoline J	Antidiabetic	[125]
*S. apiospermum*	Scequinadoline I	ND	[125]
*S. apiospermum*	Fumiquinazolines	ND	[125]

ND—not determined.

### 3.5. Molecules Involved in Biodegradation

Filamentous fungi can usually grow in xenobiotic-contaminated environments due to the production of extracellular enzymes (e.g., esterases) able to hydrolyze/oxidize the toxic compounds, transforming those molecules into intermediate metabolites, which can be further absorbed and metabolized by fungal cells [151]. In this context, *Scedosporium* species have gained attention from the scientific community due to their ability to thrive in decayed wood, manure, soils and heavily contaminated water, besides being able to adapt to high salt concentrations and low oxygen levels [153,154]. In 1968, the ability of *S. boydii* (formerly *Allescheria boydii*) to utilize *n*-alkanes (C-10; C-14; C-16 and C-18) and 1-alkenes (C-10:1; C-14:1 and C-16:1) as their only carbon source was described [155]. Subsequently, the ability of *S. boydii* strains isolated from oil-soaked soil in Canada to degrade linear aliphatic compounds was demonstrated [154]. Similarly, *S. apiospermum* is capable of utilizing aromatic compounds (diaryl ester phenylbenzoate, phenol, *p*-cresol, *p*-tolylbenzoate, 4-chlorophenylbenzoate and toluene) as the sole carbon source [156,157,158,159]. Other molecules/compounds degraded by *Scedosporium* species were reported: 2,3,7,8-tetrachlorodibenzeno-*p*-dioxin, polychlorinated biphenyl, acetaminophen, biodiesel, esters, diesel hydrocarbons, tetrahydrofuran and azo dyes (e.g., Reactive Yellow 145 and Remazol Yellow RR) (Table 2) [84,160,161,162,163,164,165,166,167].

A genome sequencing of the *S. apiospermum* environmental strain identified metabolic pathways that can potentially degrade ethylbenzene, xylene, dioxin, atrazine, styrene, naphthalene, fluorobenzoate, geraniol, chloroalkane, chloroalkene, benzoate, caprolactam, butanoate and aminobenzoate, among other hydrocarbons [168,169]. The proposed pathway of *p*-cresol degradation starts with its oxidation leading to 4-hydroxybenzylalcohol, which is converted into 4-hydroxybenzaldehyde and 4-hydroxybenzoic acid. The 4-hydroxybenzoic acid is converted into protocatechuate, which is metabolized in 3-carboxy-cis into cis-muconate, which is converted into 3-carboxymuconolactone and 3-oxoadipate [6,156]. However, some enzymes proposed in this pathway, such as hydroquinone hydroxylase, 4-hydroxybenzoate 3-hydroxylase, hydroxyquinone 1,2 dioxygenase, protocatechuate 3,4 dioxygenase and maleylacetate reductase, were not found in a genomic study of *S. apiospermum* [168], indicating that these enzymes are classified among the hypothetical, with unknown function, or they are in the gap regions of the genome [168]. Rougeron and co-workers also suggested that the conversion of 4-hydroxybanzoate into gentisate, which is cleaved into maleylpyruvate, should be considered, as all genes necessary for gentisate catabolism are organized in cluster in the *Scedosporium* genome [6]. The phenol could be catabolized either by a catechol 1,2-dioxygenase or a phenol hydroxylase, leading both pathways to a 3-oxoadipate, which can enter the tricarboxylic acid cycle [6,156]. For phenylbenzoate, the catabolism pathway initiates with the hydrolysis of diaryl ester by an esterase leading to phenol and benzoic acid [6,157]. The 4-chlorophenylbenzoate is hydrolyzed into 4-chlorophenol and benzoate, whereas the *p*-tolylbenzoate is hydrolyzed into *p*-cresol and benzoic acid [6,157].

The fungi pathway of lignin degradation starts with its extracellular oxidative degradation, which produces a mixture of aromatic monomers that are then catabolized into the upper and lower pathways. In the upper pathways, the aromatic compounds are catabolized into hydroxyquinol, catechol, protocatechuate, gentisic acid, hydroxyquinone, gallic acid and pyrogallol; subsequently, these aromatic compounds suffer a ring-opening by dioxygenases, producing degradation products that can enter the tricarboxylic acid cycle [169]. In *Scedosporium*, the putative dioxygenases gentisate 1,2-dioxygenases, homogentisate 1,2-dioxygenases, hydroxyquinol 1,2-dioxygenases, catechol 1,2-dioxygenase and protocatechuate 3,4-dioxygenase were identified through sequence homology and bioinformatic analysis, suggesting the ability of *Scedosporium* to catabolize gentisic acid, hydroxyquinol, protocatechuate and catechol [169].

A study conducted by Janda-Ulfig and co-workers [170] suggested that oil-contaminated environments could favor the growth of *Scedosporium* species, since rapeseed oil and biodiesel oil stimulate *S. boydii* growth. This could explain the most frequent occurrence of *Scedosporium* in urban, industrial and agricultural areas rather than in environments with low human activity [5,168]. In this way, three patents have been deposited for the use of *Scedosporium* species: (i) as composting promoters, (ii) in the bioremediation of nutrient-rich effluents and (iii) in the bioremediation of livestock manure [6,171,172,173,174].

## 4. Conclusions

As summarized in the present study, the emergent and multidrug-resistant *Scedosporium* species are able to secrete a vast array of distinct molecules with a wide range of biological functions, including polysaccharides, non-peptide small-molecule metabolites, non-ribosomal peptides and (glyco)protein-nature molecules. The secreted molecules can act for the benefit of human kind, such as in the bioremediation of polluted environments and in the treatment of microbial infections. However, the molecules secreted by *Scedosporium* species are also essential for its pathogenesis during human infections, such as proteolytic enzymes. In this way, studies concerning these secreted molecules can help in the development of improved diagnosis techniques and treatment of scedosporiosis.

## Figures and Tables

**Figure 1 jof-08-01172-f001:**
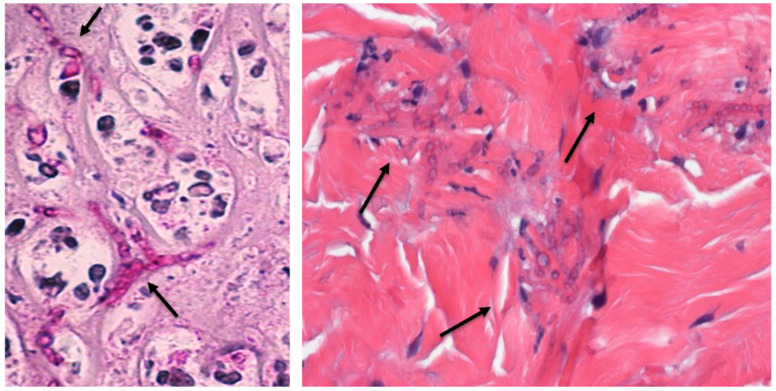
Histopathological sections evidencing *Scedosporium* fungal particles in human tissue. **Left image**: periodic acid-Schiff stain evidencing thin irregular, septate hyphae of *S. boydii* (black arrow) on a background of necrotic detritus and neutrophils in eviscerated ocular tissue (kindly donated by Dr. Virginia Vanzzini-Zago and Dr. Abelardo Rodrıguez-Reyes at Hospital Asociacion para Evitar la Ceguera en Mexico and Dr. Sonia Corredor-Casas at Instituto Mexicano de Oftalmologıa IAP Queretaro, Mexico). **Right image**: hematoxylin and eosin stain showing many hyphae of *Scedosporium* in the dermis of a skin biopsy (kindly provided by Dr. Stacy Beal, Assistant Professor, University of Florida, College of Medicine, Department of Pathology, Immunology and Laboratory Medicine, Gainesville, FL, USA). Original magnification of the images is 400×.

**Figure 2 jof-08-01172-f002:**
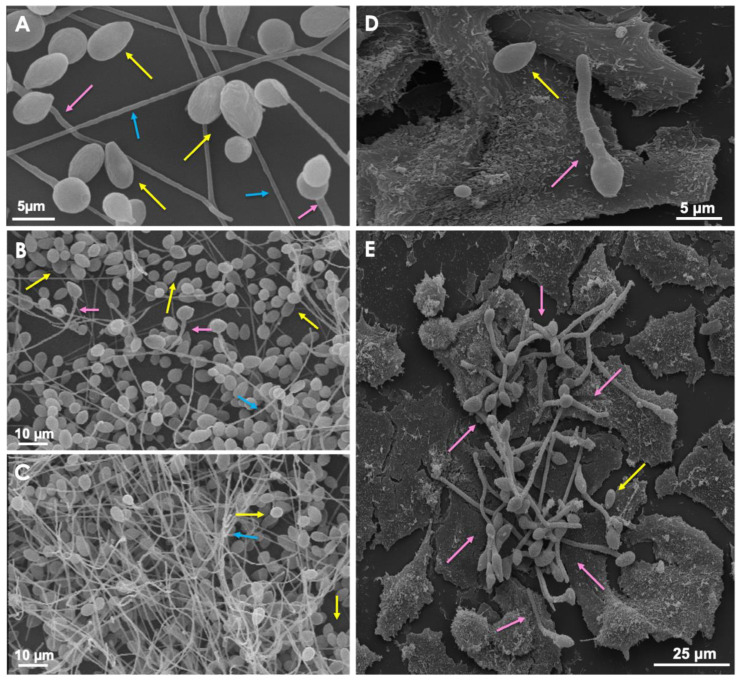
Distinct morphologies of *Scedosporium* evidenced by scanning electron microscopy. The images demonstrate the conidia (yellow arrow), germinated conidia (pink arrow) and hyphae (blue arrow) of *S. apiospermum* on glass substrate. Note in the micrographs (**A**–**C**) the mycelia formed by *S. apiospermum* on glass surface. (**D,E**) Interaction of *S. apiospermum* with A549 epithelial cells with conidia, germinated conidia and the filamentous network of hyphae cells on top of A549 cells.

**Figure 3 jof-08-01172-f003:**
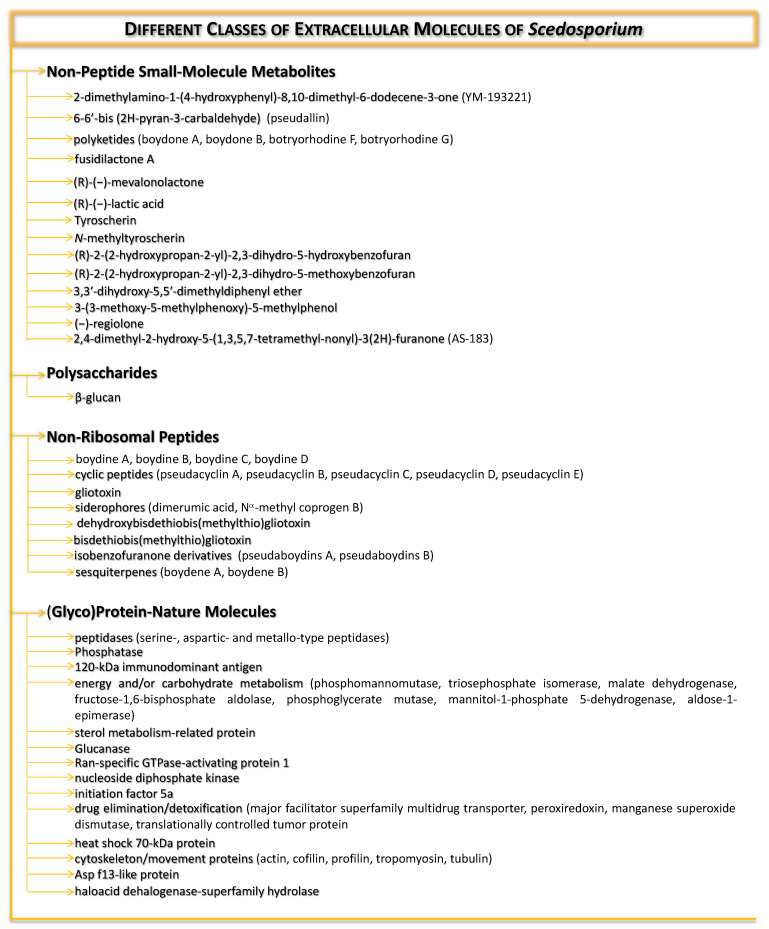
Overview of the different classes of molecules secreted/released by environmental and/or clinical strains of *Scedosporium* species.

**Figure 4 jof-08-01172-f004:**
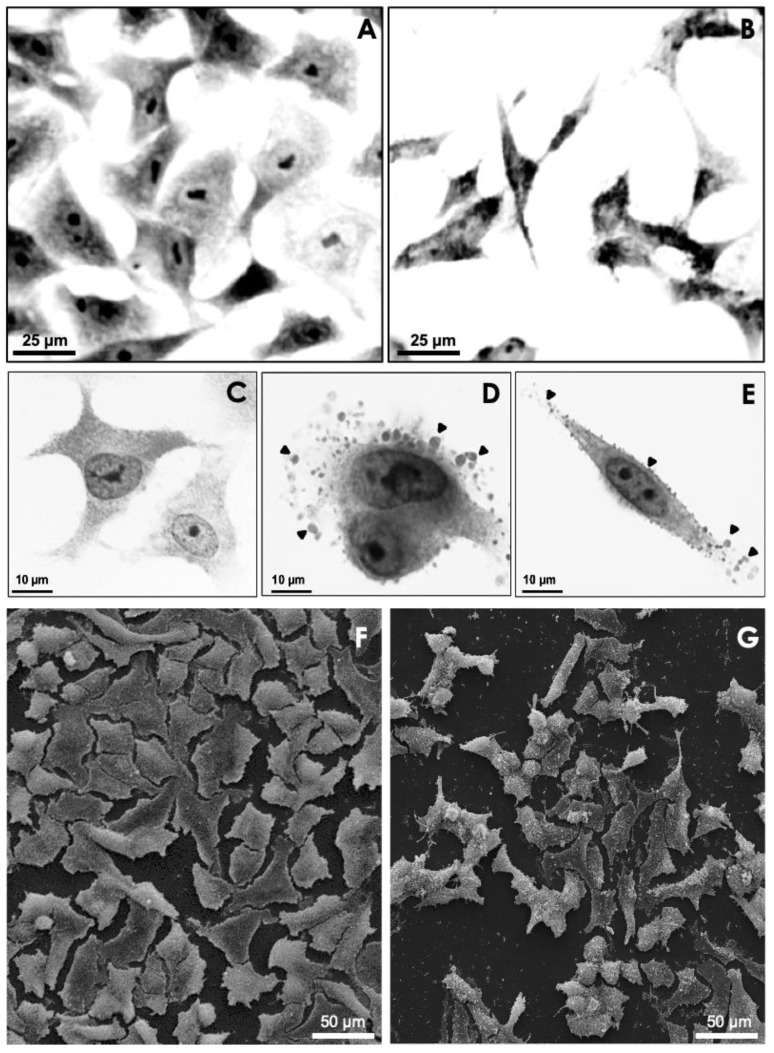
Light microscopy (**A**–**E**) and scanning electron microscopy (SEM) (**F**,**G**) images representing the monolayer of A549 epithelial cells before (**A**,**C**,**F**) and after (**B**,**D**,**E**,**G**) incubation with secreted molecules from *S. apiospermum* mycelial cells. Note the morphological changes in A549 cells, such as the release of the monolayer (**B**,**G**) and the presence of bubbles (arrowheads in **D**,**E**) on the epithelial cell surface. (Adapted from Silva et al., 2012 [35]).

**Figure 5 jof-08-01172-f005:**
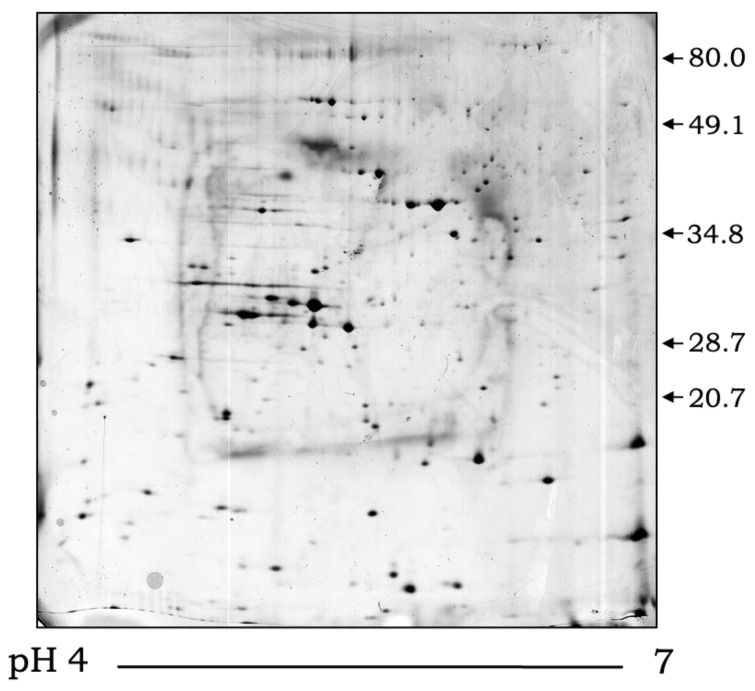
The 2-dimensional SDS-PAGE stained with Coomassie Brilliant Blue G-250 demonstrating the diversity of proteins extracellularly released by *S. apiospermum*. Molecular mass markers expressed in kilodaltons are expressed on the right.

**Figure 7 jof-08-01172-f007:**
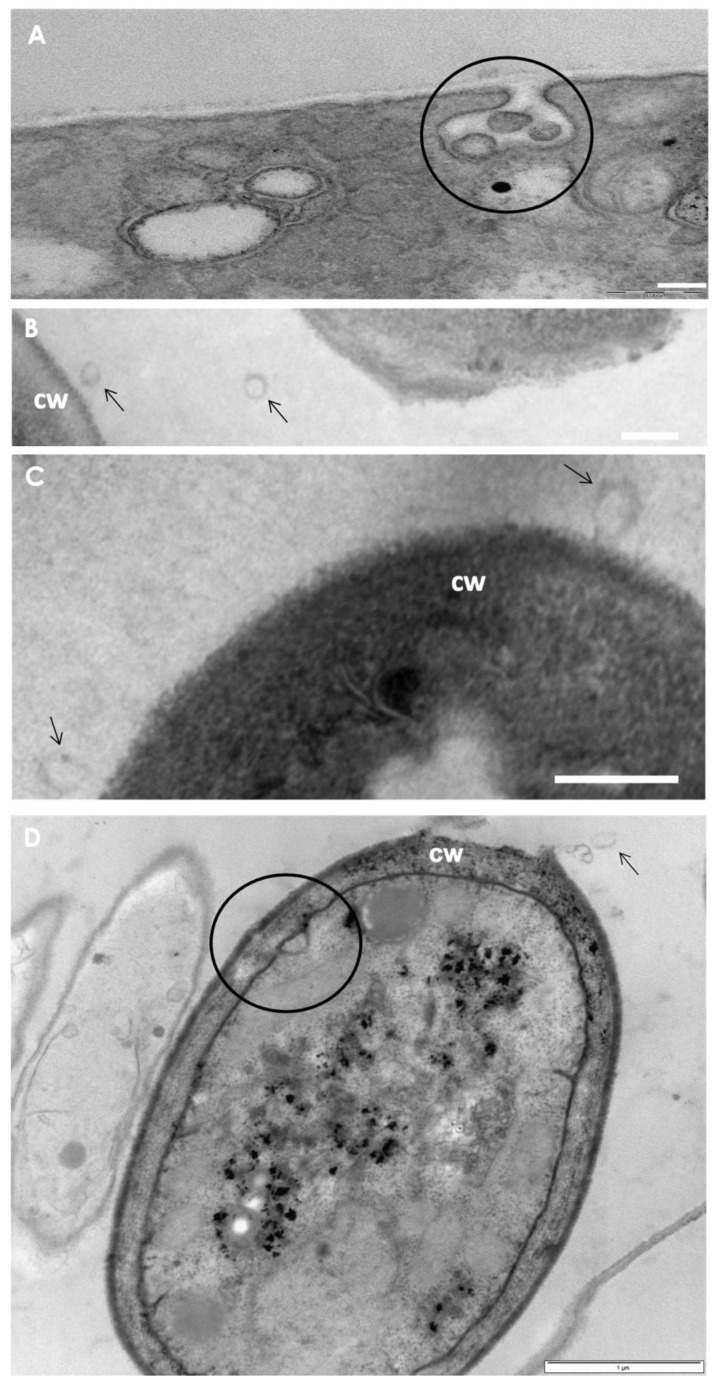
Transmission electron microscopy (TEM) showing vesicle-like structures in *Scedosporium* spp. (**A**) TEM images demonstrated the presence of vesicles within a membrane invagination in *S. minutisporum* hyphae. The black circle evidences the invagination. Bar: 1 µm. (**B**,**C**) TEM micrographs showing vesicle bodies in the extracellular milieu of *S. apiospermum* hyphae. Arrows evidence the vesicular bodies. cw, cell wall. Bar: 250 nm. (**D**) TEM image of *S. apiospermum* conidia. The black circle evidences the membrane invagination, and the arrow evidences the vesicular bodies. Cw, cell wall. Bar: 1 µm.

**Figure 8 jof-08-01172-f008:**
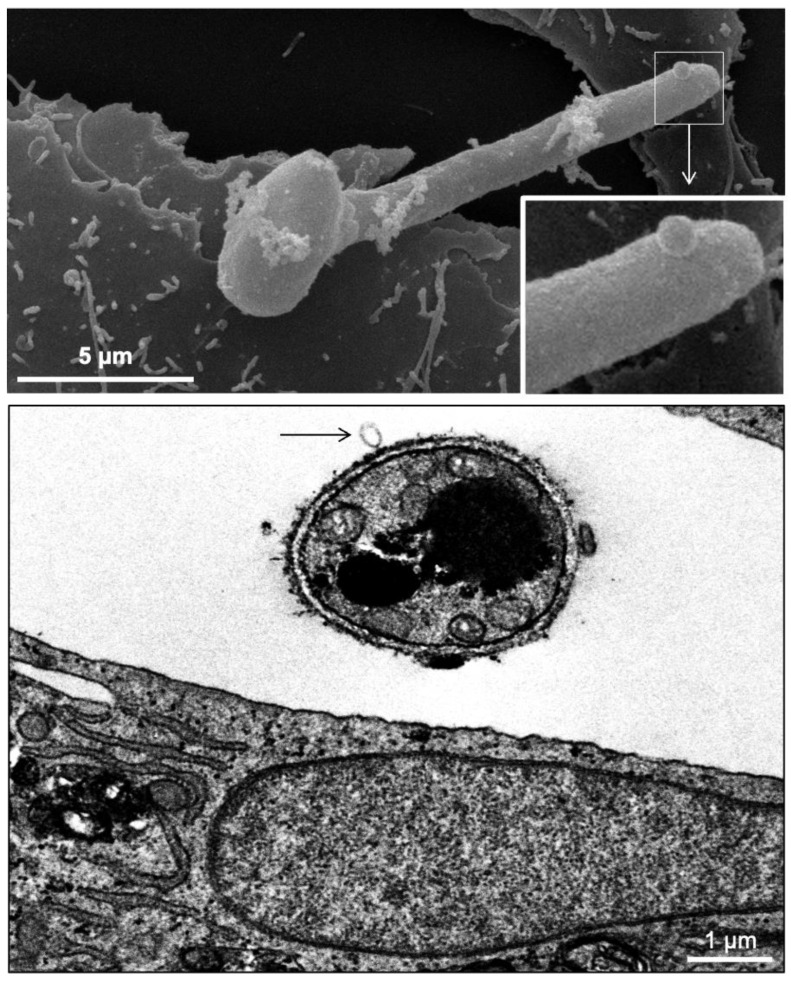
Scanning electron microscopy (SEM) demonstrating the presence of vesicles on the tip of the germinated conidia of *S. apiospermum* during interaction with A549 epithelial cells. Transmission electron microscopy (TEM) evidences (black arrow) a vesicle secreted by *S. apiospermum* in the presence of A549 epithelial cells.

**Figure 9 jof-08-01172-f009:**
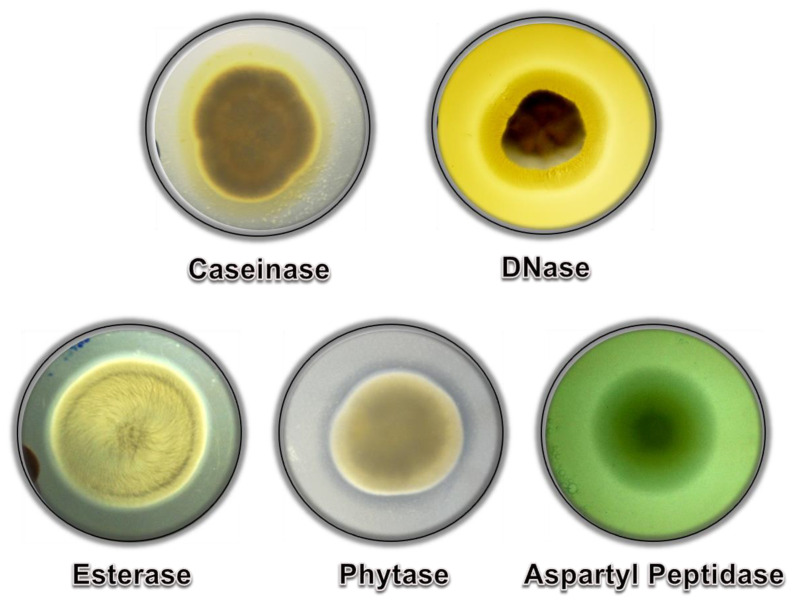
Examples of hydrolytic enzymes produced by *S. apiospermum*. The positive hydrolytic activity of caseinase, DNAse, esterase, phytase and aspartyl protease was determined through the detection of degradation halos around the fungal colony using distinct growth media containing specific substrates.

**Figure 10 jof-08-01172-f010:**
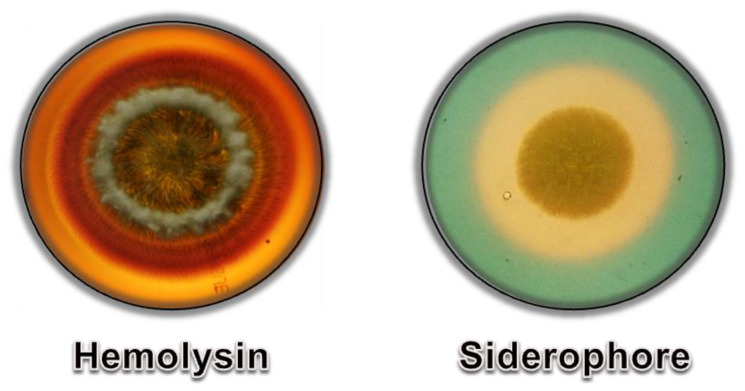
Production of hemolysin and siderophores by *S. apiospermum*. **Left**: Plate of Sabouraud medium supplemented with 7% sheep blood presenting a 7-day growth of *S. apiospermum* surrounded by a halo indicating hemolytic activity. **Right**: Plate of Sabouraud medium supplemented with chrome azurol S (CAS) and iron III solution presenting a 7-day growth of *S. apiospermum* surrounded by a halo indicating iron chelation.

**Figure 11 jof-08-01172-f011:**
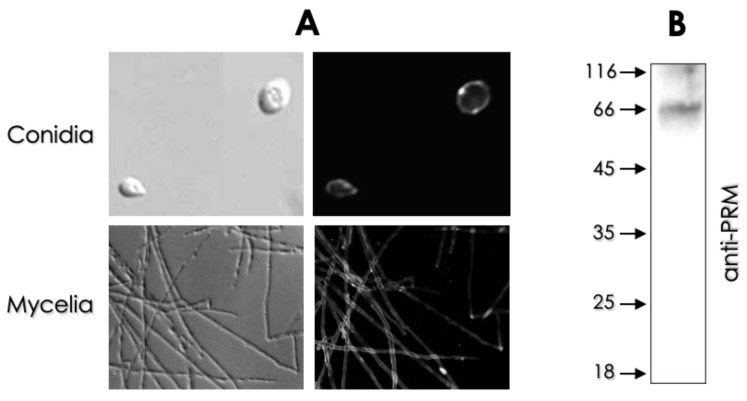
Detection of PRM produced by *S. apiospermum.* (**A**) Immunofluorescence microscopy evidencing PRM on the surface of both conidia and mycelia of *S. apiospermum* utilizing an anti-PRM antibody (kindly donated by Dr. Eliana Barreto Bergter from Instituto de Microbiologia Paulo de Góes and Universidade Federal do Rio de Janeiro). (**B**) Western blotting assay evidencing the presence of soluble PRM in the supernatant of *S. apiospermum* mycelial cells growth using anti-PRM antibody.

**Figure 12 jof-08-01172-f012:**
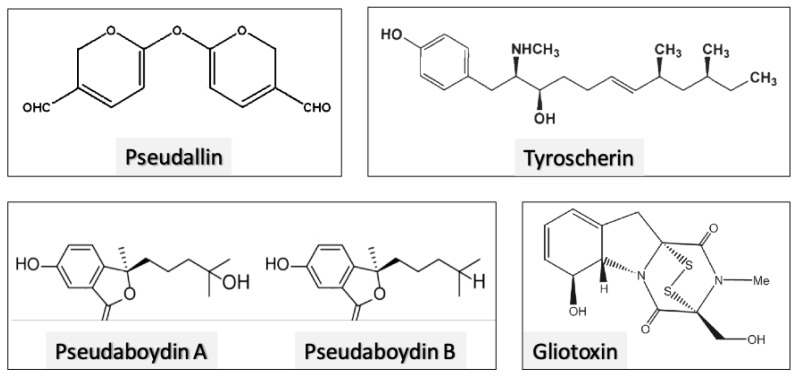
Some examples of non-peptide small-molecule metabolites extracellularly released by environmental and clinical strains of *Scedosporium* spp.

**Table 2 jof-08-01172-t002:** List of xenobiotics degraded by *Scedosporium* species in the literature.

Species	Molecule/Compound	References
*S. boydii*	*n*-Alkanes	[155]
*S. boydii*	1-Alkenes	[155]
*S. boydii*	Hydrocarbons	[155,165]
*S. apiospermum*	Phenol	[156]
*S. apiospermum*	*p*-Cresol	[156]
*S. apiospermum*	4-Hydroxybenzoate	[156]
*S. apiospermum*	4-Hydroxybenzaldehyde	[156]
*S. apiospermum*	4-Hydroxybenzylalcohol	[156]
*S. apiospermum*	Protocatechuate	[156]
*S. apiospermum*	Phenylbenzoate	[157]
*S. apiospermum*	*p*-Tolylbenzoate	[157]
*S. apiospermum*	4-Chlorophenylbenzoate	[157]
*S. apiospermum*	Toluene	[158]
*S. boydii*	Rapeseed oil	[168]
*S. boydii*	Biodiesel	[83,84,168]
*S. boydii*	Diesel oil	[84,168]
*S. boydii*	2,3,7,8-Tetrachlorodibenzo-*p*-dioxin	[160]
*S. apiospermum*	Polychlorinated biphenyl	[161]
*S. apiospermum*	Polycyclic aromatic hydrocarbons	[159]
*S. dehoogii*	Acetaminophen	[162]
*S. boydii*	Diesel blend	[84]
*S. boydii*	Tetrahydrofuran	[163]
*S. boydii*	Dibutyl tin dilaurate	[164]
*S. boydii*	Di-*n*-butyl-oxo-stannane	[164]
*Scedosporium* spp.	Lignin	[169]
*S. apiospermum*	Olive mill wastewater	[166]
*S. apiospermum*	Reactive Yellow 145	[167]
*S. apiospermum*	Remazol Yellow RR	[167]

## Data Availability

Not applicable.

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
