# Peer review of "Extracellularly Released Molecules by the Multidrug-Resistant Fungal Pathogens Belonging to the *Scedosporium* Genus: An Overview Focused on Their Ecological Significance and Pathogenic Relevance"

_jof, 2022, doi:10.3390/jof8111172_

Round 1
Reviewer 1 Report
The authors have submitted a review on the fungal pathogen Scedosporium. The emphasis of the review is on classes of extracellular molecules, which is extensive in scope. Nevertheless, the manuscript is well-written and quite informative especially for those of us that know very little about this fungal pathogen. The extracellular compounds that are described appear to have important functions during the cell cycle. Most are non-peptide small molecule metabolites.
The review focuses upon presumed functions; I would encourage the authors to tell us about what is ahead for this interesting group of organisms and their extracellular molecules. Thus, the review is very descriptive. I am certain that the compounds have been assigned functions based upon their presumed activity. Less clear is functional analysis; I do not know what the status is regarding the molecular pathogenesis of virulence with these pathogens. The authors should discuss efforts by this community in sequencing the genome and providing information that speaks to molecular pathogenesis. I realize this is a big order in an already extensive manuscript. assigning genes and their functions.
Other points.
Table 3 is complex and should indicate published references to each of the compounds. In section 2, the authors discuss “specialized functions during stages of the of the infectious process.” But to me this review focuses upon presumed functions, which is fine. A summary statement on this subject should be included.
A lot of the data presented in the review seems to be unpublished, for example, Figures 5, 6, 7, 8, and 9. I do not see references in Figure legends. Figure 9 appears to be relatively non-quantifiable. Why were these molecules shown?
The review is important, to me a lot like the research published initially about roles of wall, proteases, capsule, germination, dimorphisms, etc, of other fungal pathogens. The review is important and probable my comments are a bit unfair.
As mentioned above, there should be a 1-2 paragraph section in the review of the state of molecular pathogenesis with this fungus.
The EM provides useful information that can be useful in establishing identity of components in the vesicles pictured. I do not expect that data be included. Perhaps tell us what the next step is.
Figure 5 data demonstrate that the 2-dimensional analysis methodology is in good shape. The authors state that these experiments are “being redone.”
The review is long, but I think necessary for stirring up the interest in this group of pathogens. Frankly the only suggestion I can make is to try and shorten section 3.
Author Response
Thank you for your message from October 7th about our manuscript “Extracellularly released molecules by the multidrug-resistant fungal pathogens belonging to the Scedosporium genus: an overview focused on their ecological significance and pathogenic relevance” (jof-1917550), submitted for publication in Journal of Fungi. We would like to thank to the reviewers for the critical comments. In this regard, I would like to express our position on each point raised by the reviewers, as described below.
Authors’ Comments to the Reviewer 1
Reviewer 1: The authors have submitted a review on the fungal pathogen Scedosporium. The emphasis of the review is on classes of extracellular molecules, which is extensive in scope. Nevertheless, the manuscript is well-written and quite informative especially for those of us that know very little about this fungal pathogen. The extracellular compounds that are described appear to have important functions during the cell cycle. Most are non-peptide small molecule metabolites.
Authors: Firstly, the authors are grateful for the reviewer’s kind comments and valuable suggestions on the manuscript.
Reviewer 1: The review focuses upon presumed functions; I would encourage the authors to tell us about what is ahead for this interesting group of organisms and their extracellular molecules. Thus, the review is very descriptive. I am certain that the compounds have been assigned functions based upon their presumed activity. Less clear is functional analysis; I do not know what the status is regarding the molecular pathogenesis of virulence with these pathogens. The authors should discuss efforts by this community in sequencing the genome and providing information that speaks to molecular pathogenesis. I realize this is a big order in an already extensive manuscript. assigning genes and their functions.
Authors: The knowledge about this fungal genus is very incipient. In fact, very little is reported until now about the pathogenesis of scedosporiosis as well as the fungal molecules (directly/indirectly) involved in the course of the infection. The review compilates all the reported molecules produced by Scedosporium spp. with known functions on this theme as well as other secreted molecules which the function was not yet reported but that are well-known molecules involved in pathogenesis considering other clinically relevant fungi. All the information’s regarding the secreted molecules by Scedosporium species were offered and, unhappily, the lapse in the knowledge exists. For this reason, the authors decided to write this review in order to contribute to the literature, helping to understand potential extracellularly released molecules with potential role(s) in the pathogenesis. Just to corroborate our statements, the genome sequence of Scedosporium (the first one was S. apiospermum) was published in 2014, but until now we do not have much information about the molecules involved in the pathogenesis of these intriguing filamentous fungi (Vandeputte et al., 2014). The authors wish that the present review stimulates new scientists to study these intriguing, emergent, multidrug-resistant filamentous fungi.
Reviewer 1: Table 3 is complex and should indicate published references to each of the compounds.
Authors: As there is not a Table 3 in the manuscript, the authors do believe that the reviewer intended to make mention of Figure 3. In this sense, the references to each compound were added in the Figure 3 as requested by the reviewer.
Reviewer 1: In section 2, the authors discuss “specialized functions during stages of the of the infectious process.” But to me this review focuses upon presumed functions, which is fine. A summary statement on this subject should be included.
Authors: The reviewer cited a passage on the text as the full text is “Throughout evolution, some fungi have adapted the secretion machinery in order to perform a great number of specialized functions during different stages of the infectious process, allowing these microorganisms to cause illness.” In fact, the sentence translates the ability and necessity of all fungal cells to secrete molecules for their survival, since fungi need to release molecules to cleave and obtain their nutrients since the extracellular digestion is imperative due to the fungal cell wall, which is a physical barrier to large substrates. So, this is not a presumed function, in fact, it is a well-known process developed for all known fungal cells.
Reviewer 1: A lot of the data presented in the review seems to be unpublished, for example, Figures 5, 6, 7, 8, and 9. I do not see references in Figure legends. Figure 9 appears to be relatively non-quantifiable. Why were these molecules shown?
Authors: Except for figure 6, which is a new result, all the other figures correspond to data published in previous papers of our group. However, all the figures (as a whole) are composed of new and inedited images. For all these reasons, the authors did not add a reference in the figure legend, since there are new images reporting results of previous works that were cited in the regular text. To finalize, the unpublished data as well as new images regarding previous published papers from the author’s laboratory were added to the review in order to enrich the review content and to make the review attractive to the readers. The figure 9 was added just to demonstrate that Scedosporium cells are able to release a diverse number of extracellular enzymes by using solid medium incorporated with different substrates. In fact, it is possible to quantify the activity by measuring the Pz value (which means the ratio of colony diameter and the diameter of colony plus precipitation zone measured by a paquimeter), but it is not did because the idea was only shown the presence of the degradation halo around the fungal colony.
Reviewer 1: The review is important, to me a lot like the research published initially about roles of wall, proteases, capsule, germination, dimorphisms, etc, of other fungal pathogens. The review is important and probable my comments are a bit unfair.
Authors: The authors thank the reviewer again for the generous comments.
Reviewer 1: As mentioned above, there should be a 1-2 paragraph section in the review of the state of molecular pathogenesis with this fungus.
Authors: As said earlier, the molecular pathogenesis of Scedosporium infections is yet unknown. However, along all the text and in different moments, the authors highlighted the proposed role of some individual fungal molecules in the pathogenesis process, like proteases, lipases, and many others.
Reviewer 1: The EM provides useful information that can be useful in establishing identity of components in the vesicles pictured. I do not expect that data be included. Perhaps tell us what the next step is.
Authors: A more complete study containing unpublished data about the vesicles secreted by S. apiospermum is being prepared by our research group and as soon as possible will be sent to publication. In this new publication, the authors isolated the extracellular vesicles secreted by S. apiospermum and we characterized the main molecules with emphasis in protein-forming these intriguing structures as well as we performed some experiments to clarify the potential role of these vesicles on in vitro mammalian cells and in vivo using the Galleria mellonella larvae.
Reviewer 1: Figure 5 data demonstrate that the 2-dimensional analysis methodology is in good shape. The authors state that these experiments are “being redone.”
Authors: The 2D gels were already published and, in this sense, the 2D-secreted protein map is previously published together with the proteomic analysis of the main 25 spots (Silva et al., 2012 doi: 10.1021/pr200875x). In that previous work the genome of the Scedosporium is not yet known, which generated several problems in the protein identification. Based on this statement, the 2-dimensional SDS-PAGE was redonned, because we have now the genome sequencing of S. apiospermum (Vandeputte et al., 2014 doi: 10.1128/genomeA.00988-14), and we finished the new experiments in order to reanalyzed the secretome by different approaches in order to get a more realist profile regarding the proteins secreted by this fungus.
Reviewer 1: The review is long, but I think necessary for stirring up the interest in this group of pathogens. Frankly the only suggestion I can make is to try and shorten section 3.
Authors: The authors thank the reviewer again for the kind comments. Regarding the section 3, the authors implore the reviewer to maintain the section as it stays, since it is the part that summarizes the main focus of this review. The authors also do believe that the data are relevant, particularly due to the scarce reviews on the theme. The authors hope that the reviewer understands our position on this regard.
Reviewer 2 Report
In this paper, the authors review the molecules secreted by Scedosporium species in their environment. This review is interesting, but it does not seem really suitable for publication in this special issue dedicated to Fungi and Fungal Metabolites for the Improvement of Human and Animal Life, Nutrition and Health. Most of the molecules discussed in the paper have no potential for improvement of human and animal life, nutrition, and health. In contrast, they help these fungi to multiply within the host and to evade the host immune response, thus allowing the fungi to cause infections and to damage human or animal health.
It is thus suggested to focus this review: i) on the potential use of Scedosporium species as bioremediation agents by the degradation of aliphatic or aromatic pollutants, which is the only process with a real potential for improvement of human and animal life and health; and ii) on secreted secondary metabolites with known biological activities.
Alternatively, it is suggested to publish this review in another special issue of Journal of Fungi which is currently being prepared, focused on Scedosporium and Lomentospora species, where the present review will be better suited.
Specific comments
Line 17: All fungi are not ubiquitous organisms.
Lines 36-40: in the consensus review published by the ECMM/ISHAM working group on Scedosporium/Lomentospora infections (Ramirez-Garcia et al, Med Mycol, 2018), only ten species have been recognized (i.e. S. angustum, S. apiospermum, S. aurantiacum, S. boydii, S. cereisporum, S. dehoogii, S. desertorum, S. ellipsoideum, S. fusoideum and S. minutisporum). Regarding the two other species cited here, S. rarisporum and S. sanyaense described in 2017 by Han et al. in Mycosystema, they were only cited in reference 7 and further studies are needed for definitive validation.
Lines 40-42: regarding reassignment of Scedosporium prolificans to the genus Lomentospora, please change reference 8 for the community paper from the ECMM/ISHAM working group on Scedosporium/Lomentospora infections: Lackner M, de Hoog SG, Yang L, Ferreira Moreno L, Ahmed SA, Andreas F, Kaltseis J, Nagl M, Lass-Flörl C, Risslegger B, Rambach G, Speth C, Robert V, Buzina W, Chen S, Bouchara JP, Cano-Lira JF, Guarro J, Gené J, Fernández Silva F, Haido R, Haase G, Havlicek V, Garcia-Hermoso D, Meis JF, Hagen F, Kirchmair M, Rainer J, Schwabenbauer K, Zoderer M, Meyer W, Gilgado F, Vicente VA, Piecková E, Regenermel M, Rath PM, Steinmann J, Wellington de Alencar X, Symoens F, Tintelnot K, Ulfig K, Velegraki A, Tortorano AM, Giraud S, Mina S, Rigler-Hohenwarter K, Hernando F, Ramirez-Garcia A, Pellón A, Kaur J, Barreto Bergter E, Vieira de Meirelles J, Dutra da Silva I, Delhaes L, Alastruey-Izquerdo A, Li RY, Lu Q, Moussa T, Almaghrabi O, Al-Zahrani H, Okada G, Deng S, Liao W, Zeng J, Issakainen J, Liporagi Lopes LC. Proposed nomenclature for Pseudallescheria, Scedosporium and related genera. Fungal Diversity, 2014, 67 : 1-10.
Lines 46-48: Disseminated infections do not occur in immunocompetent patients.
Lines 52-55: This sentence refers only to disseminated scedosporiosis.
Lines 59-61: A disseminated infection was not found by Husain et al. in 69% and 46% of HSCT and SOT recipients, but the infection was disseminated in 69% and 46% of HSCT and SOT recipients with a Scedosporium infection, which is not strictly the same.
Line 62: Delete in disseminated disease since this is already said at the beginning of the sentence.
Line 77: Scedosporium species rank second among the filamentous fungi most frequently isolated from CF patients.
Lines 100-104: Not all the fungal components allowing the success of the host tissue colonization are secreted.
Figure 2: No obvious biofilm on Figure 2 B and C.
Figure 3: Not fully readable because of a problem when preparing the pdf file. Nevertheless, there are some mistakes in this figure since some of the components called non-peptide small-molecule metabolites are in fact produced through non ribosomal peptide synthases and therefore should be cited with non-ribosomal peptides (siderophores, gliotoxin and its derivatives, aranotin derivatives, and components called epipolythiodioxopiperazines). Please note that gliotoxin and its derivatives as well as aranotin derivatives are also epipolythiodioxopiperazines.
Line 145: due to their ability
Line 184: the genome sequence of S. apiospermum is now available
Line 195: viable or valuable alternative model?
Lines 201-203: Not useful, since it is the definition of a dose-dependent effect.
Lines 196-205: Not really a supernatant. As culture supernatants were concentrated by dialysis using a 10-kDa Amicon membrane, all small molecules that have released by the fungus during its growth have been lost during the dialysis. With this methodology, only the effects of secreted proteins were investigated.
Lines 242-245: yeasts of … Alternaria infectoria. This filamentous fungus is not known to be dimorphic.
Line 395: Not only the presence of catalase A1 was evaluated in the extracellular milieu, but this enzyme was also purified to homogeneity and characterized by Mina et al. (96).
Lines 398-399: A peroxiredoxin was also detected on the secretome of S. apiospermum [35], but its role on Scedosporium is poorly known. This sentence should be modified, as it was demonstrated in a transcriptomic analysis of the enzymatic antioxydants produced by S. apiospermum, that Saprx2 and SAPIO_CDS1830 encoding, respectively, one of the three peroxiredoxins and one the two thioredoxin reductases identified in the fungal genome, were overexpressed in response to all the stress conditions studies, including co-cultivation with phagocytic cells. See Transcriptional profiling of Scedosporium apiospermum enzymatic antioxidant gene battery unravels the involvement of thioredoxin reductases against chemical and phagocytic cells oxidative stress. Med Mycol, 2019, 57(3) : 363-373.
Lines 432-434: why most ineffective? This may be the case only with polymicrobial clinical samples, like sputum samples from patients with cystic fibrosis if a Scedosporium-selective culture medium is not inoculated in parallel to Sabouraud dextrose agar with antibiotics. On the contrary, in case of disseminated infections, Scedosporium and Lomentospora species can be detected from blood samples conversely to Aspergillus fumigatus, which has never been detected in automated blood cultures.
Lines 456-464: No relation with improvement of the diagnosis of a Scedosporium infection. Delete the corresponding sentences.
Same comment for lines 473-480.
Table 1 should focus on secreted secondary metabolites with known biological activities. For example, brefeldin A, ergosterol, ergosterol peroxide, cerevisterol and ducitol have been identified from a mycelial extract, and they have never been demonstrated to be secreted by Scedosporium species. Likewise, a biological activity has not been demonstrated for many of the secondary metabolites listed in this table.
Section 3.5: More than a list of pollutants that may be degraded by Scedosporium species, it would have been interesting to present the degradation pathways and the enzymes involved. In addition, patents on the use of Scedosporium species as bioremediation agents should be presented since they are directly in the focus of this special issue.
Berthon JY, Grizard D. Fungal inoculum, method for its preparation and methods for its utilisation for the treatment of waste water with high content in organic material. EP1352953. 2003.
Nomoto T. Promoter for composting. JP3485345. 2003.
Laugero C, Tillier D. Method for biological treatment of animal breeding effluent and device therefor. EP1242318. 2008.
Author Response
Thank you for your message from October 7th about our manuscript “Extracellularly released molecules by the multidrug-resistant fungal pathogens belonging to the Scedosporium genus: an overview focused on their ecological significance and pathogenic relevance” (jof-1917550), submitted for publication in Journal of Fungi. We would like to thank to the reviewers for the critical comments. In this regard, I would like to express our position on each point raised by the reviewers, as described below.
Authors’ Comments to the Reviewer 2
Reviewer 2: In this paper, the authors review the molecules secreted by Scedosporium species in their environment. This review is interesting, but it does not seem really suitable for publication in this special issue dedicated to Fungi and Fungal Metabolites for the Improvement of Human and Animal Life, Nutrition and Health. Most of the molecules discussed in the paper have no potential for improvement of human and animal life, nutrition, and health. In contrast, they help these fungi to multiply within the host and to evade the host immune response, thus allowing the fungi to cause infections and to damage human or animal health. It is thus suggested to focus this review: i) on the potential use of Scedosporium species as bioremediation agents by the degradation of aliphatic or aromatic pollutants, which is the only process with a real potential for improvement of human and animal life and health; and ii) on secreted secondary metabolites with known biological activities. Alternatively, it is suggested to publish this review in another special issue of Journal of Fungi which is currently being prepared, focused on Scedosporium and Lomentospora species, where the present review will be better suited.
Authors: The authors respectfully disagree with the reviewer, as the authors think that the topics addressed in the review are yes about the improvement of human life and health. This can be reflected in the topics about the virulence factors, which the studies can improve the clinical management of patients, as also the use of molecules secreted by the fungi for a better diagnosis. In addition, the use of fungal metabolites in degradation of pollutants as already appointed by the reviewer. In addition, the theme of the present review was previously approved by the editors responsible for this special issue and a waiver was kindly reserved by us, of course, if the manuscript is acceptable for publication. In this context, the Brazilian authors are extremely beneath with this kind of action which should be highlighted, particularly in the present days in which the Brazilian government drastically reduced the investments in both education and science. So, the authors generously ask the reviewer to change its position regarding all these raised critical points. Also, the authors wish the reviewer understands our comments and appeal.
Reviewer 2: Line 17: All fungi are not ubiquitous organisms.
Authors: The authors add the word “some” before “fungi” in order to specify that only some species are ubiquitous.
Reviewer 2: Lines 36-40: in the consensus review published by the ECMM/ISHAM working group on Scedosporium/Lomentospora infections (Ramirez-Garcia et al, Med Mycol, 2018), only ten species have been recognized (i.e. S. angustum, S. apiospermum, S. aurantiacum, S. boydii, S. cereisporum, S. dehoogii, S. desertorum, S. ellipsoideum, S. fusoideum and S. minutisporum). Regarding the two other species cited here, S. rarisporum and S. sanyaense described in 2017 by Han et al. in Mycosystema, they were only cited in reference 7 and further studies are needed for definitive validation.
Authors: The species S. rarisporum and S. sanyaense were removed from the text.
Reviewer 2: Lines 40-42: regarding reassignment of Scedosporium prolificans to the genus Lomentospora, please change reference 8 for the community paper from the ECMM/ISHAM working group on Scedosporium/Lomentospora infections: Lackner M, de Hoog SG, Yang L, Ferreira Moreno L, Ahmed SA, Andreas F, Kaltseis J, Nagl M, Lass-Flörl C, Risslegger B, Rambach G, Speth C, Robert V, Buzina W, Chen S, Bouchara JP, Cano-Lira JF, Guarro J, Gené J, Fernández Silva F, Haido R, Haase G, Havlicek V, Garcia-Hermoso D, Meis JF, Hagen F, Kirchmair M, Rainer J, Schwabenbauer K, Zoderer M, Meyer W, Gilgado F, Vicente VA, Piecková E, Regenermel M, Rath PM, Steinmann J, Wellington de Alencar X, Symoens F, Tintelnot K, Ulfig K, Velegraki A, Tortorano AM, Giraud S, Mina S, Rigler-Hohenwarter K, Hernando F, Ramirez-Garcia A, Pellón A, Kaur J, Barreto Bergter E, Vieira de Meirelles J, Dutra da Silva I, Delhaes L, Alastruey-Izquerdo A, Li RY, Lu Q, Moussa T, Almaghrabi O, Al-Zahrani H, Okada G, Deng S, Liao W, Zeng J, Issakainen J, Liporagi Lopes LC. Proposed nomenclature for Pseudallescheria, Scedosporium and related genera. Fungal Diversity, 2014, 67 : 1-10.
Authors: The reference was changed.
Reviewer 2: Lines 46-48: Disseminated infections do not occur in immunocompetent patients.
Authors: The sentence was corrected.
Reviewer 2: Lines 52-55: This sentence refers only to disseminated scedosporiosis.
Authors: The “disseminated” word was added to the phrase.
Reviewer 2: Lines 59-61: A disseminated infection was not found by Husain et al. in 69% and 46% of HSCT and SOT recipients, but the infection was disseminated in 69% and 46% of HSCT and SOT recipients with a Scedosporium infection, which is not strictly the same.
Authors: The information was corrected.
Reviewer 2: Line 62: Delete in disseminated disease since this is already said at the beginning of the sentence.
Authors: The sentence was corrected.
Reviewer 2: Line 77: Scedosporium species rank second among the filamentous fungi most frequently isolated from CF patients.
Authors: The sentence was corrected.
Reviewer 2: Lines 100-104: Not all the fungal components allowing the success of the host tissue colonization are secreted.
Authors: The authors substituted the term “rely on” with “is partially due” in order to clarify that some of the essential steps for tissue colonization involves secreted molecules, but not only.
Reviewer 2: Figure 2: No obvious biofilm on Figure 2 B and C.
Authors: The word “biofilm” was replaced by “mycelia”.
Reviewer 2: Figure 3: Not fully readable because of a problem when preparing the pdf file. Nevertheless, there are some mistakes in this figure since some of the components called non-peptide small-molecule metabolites are in fact produced through non ribosomal peptide synthases and therefore should be cited with non-ribosomal peptides (siderophores, gliotoxin and its derivatives, aranotin derivatives, and components called epipolythiodioxopiperazines). Please note that gliotoxin and its derivatives as well as aranotin derivatives are also epipolythiodioxopiperazines.
Authors: The errors in the figure were corrected.
Reviewer 2: Line 145: due to their ability
Authors: The sentence was corrected.
Reviewer 2: Line 184: the genome sequence of S. apiospermum is now available
Authors: The proteomic analysis regarding the proteins extracellularly released by S. apiospermum is currently being redone at the authors, since the genome sequence of S. apiospermum is available which will help to identify proteins not identified in the first study. Just to give the reviewer an idea, more than 150 proteins were revealed with the new methodology applied and using the fungal genome repertoire; the manuscript describing these new data is in progress.
Reviewer 2: Line 195: viable or valuable alternative model?
Authors: The authors meant “valuable” instead of “viable”. The sentence was corrected.
Reviewer 2: Lines 201-203: Not useful, since it is the definition of a dose-dependent effect.
Authors: The redundance was removed.
Reviewer 2: Lines 196-205: Not really a supernatant. As culture supernatants were concentrated by dialysis using a 10-kDa Amicon membrane, all small molecules that have released by the fungus during its growth have been lost during the dialysis. With this methodology, only the effects of secreted proteins were investigated.
Authors: The word “supernatant” was replaced by “secreted proteins”
Reviewer 2: Lines 242-245: yeasts of … Alternaria infectoria. This filamentous fungus is not known to be dimorphic.
Authors: The “Alternaria infectoria” after yeast organisms was removed in order to fix this error.
Reviewer 2: Line 395: Not only the presence of catalase A1 was evaluated in the extracellular milieu, but this enzyme was also purified to homogeneity and characterized by Mina et al. (96).
Authors: The presence of catalase A1 in the extracellular milieu and in the mycelial extract was described in lines 415-419.
Reviewer 2: Lines 398-399: A peroxiredoxin was also detected on the secretome of S. apiospermum [35], but its role on Scedosporium is poorly known. This sentence should be modified, as it was demonstrated in a transcriptomic analysis of the enzymatic antioxydants produced by S. apiospermum, that Saprx2 and SAPIO_CDS1830 encoding, respectively, one of the three peroxiredoxins and one the two thioredoxin reductases identified in the fungal genome, were overexpressed in response to all the stress conditions studies, including co-cultivation with phagocytic cells. See Staerck C, Tabiasco J, Godon C, Delneste Y, Bouchara JP, Fleury MJJ. Transcriptional profiling of Scedosporium apiospermum enzymatic antioxidant gene battery unravels the involvement of thioredoxin reductases against chemical and phagocytic cells oxidative stress. Med Mycol, 2019, 57(3) : 363-373.
Authors: The paragraph has been adjusted and the authors thank the reviewer for the precious contribution.
Reviewer 2: Lines 432-434: why most ineffective? This may be the case only with polymicrobial clinical samples, like sputum samples from patients with cystic fibrosis if a Scedosporium-selective culture medium is not inoculated in parallel to Sabouraud dextrose agar with antibiotics. On the contrary, in case of disseminated infections, Scedosporium and Lomentospora species can be detected from blood samples conversely to Aspergillus fumigatus, which has never been detected in automated blood cultures.
Authors: The sentence was corrected to fit the reviewer’s comment.
Reviewer 2: Lines 456-464: No relation with improvement of the diagnosis of a Scedosporium infection. Delete the corresponding sentences.
Same comment for lines 473-480.
Authors: Is true that the PRM has not been used as a diagnostic tool or it has improved the diagnosis of scedosporiosis, but the authors think that it is important to cite that this molecule could be used as a diagnostic tool as originally reported by those authors.
Reviewer 2: Table 1 should focus on secreted secondary metabolites with known biological activities. For example, brefeldin A, ergosterol, ergosterol peroxide, cerevisterol and ducitol have been identified from a mycelial extract, and they have never been demonstrated to be secreted by Scedosporium species. Likewise, a biological activity has not been demonstrated for many of the secondary metabolites listed in this table.
Authors: The authors think that is important to maintain the secondary metabolites with unknown activities in Table 1 to have a more complete overview of the already identified secondary metabolites produced by Scedosporium species. The molecules identified only in the mycelial extract were removed from the table; the authors apologize for the mistake and thank the reviewer for pointing out the error.
Reviewer 2: Section 3.5: More than a list of pollutants that may be degraded by Scedosporium species, it would have been interesting to present the degradation pathways and the enzymes involved. In addition, patents on the use of Scedosporium species as bioremediation agents should be presented since they are directly in the focus of this special issue.
Berthon JY, Grizard D. Fungal inoculum, method for its preparation and methods for its utilisation for the treatment of waste water with high content in organic material. EP1352953. 2003.
Nomoto T. Promoter for composting. JP3485345. 2003.
Laugero C, Tillier D. Method for biological treatment of animal breeding effluent and device therefor. EP1242318. 2008.
Authors: The authors are grateful for the reviewer’s valuable suggestions. In this way, the patents and degradation pathways were added to the text.
Reviewer 3 Report
In this manuscript, the authors summarized overview focused on ecological significance and pathogenic relevance of the multidrug-resistant fungal pathogens belonging to the Scedosporium genus. The content of this manuscript is well organized. This manuscript contains content that is of interest to experts in this field as well as non-experts. The manuscript has a merit to be published in Journal of Fungi. To make this manuscript even better, please consider the following comments.
1. The authors need to mention in more detail whether a review article similar to the content of this manuscript was previously published. That will be useful information for the readers of this journal.
2. The relevance of the topics covered in this review to the work of the authors to date should be mentioned in the introduction section.
3. The size of Figure 3 is too large. Also, for Non-Peptide Small-Molecule Metabolites and Non-Ribosomal Peptides in Figure 3, structural formulas of those compounds should be added.
4. The authors should add the structural formulas of the compounds listed in Table 1. That would be more beneficial to the readers of this journal.
5. The size of Figure 12 is too large. The authors should correct this.
6. If possible, the authors should be discussed the biosynthetic pathways of secondary metabolites and their relevance to pathogenicity.
Author Response
Thank you for your message from October 7th about our manuscript “Extracellularly released molecules by the multidrug-resistant fungal pathogens belonging to the Scedosporium genus: an overview focused on their ecological significance and pathogenic relevance” (jof-1917550), submitted for publication in Journal of Fungi. We would like to thank to the reviewers for the critical comments. In this regard, I would like to express our position on each point raised by the reviewers, as described below.
Authors’ Comments to the Reviewer 3
Reviewer 3: In this manuscript, the authors summarized overview focused on ecological significance and pathogenic relevance of the multidrug-resistant fungal pathogens belonging to the Scedosporium genus. The content of this manuscript is well organized. This manuscript contains content that is of interest to experts in this field as well as non-experts. The manuscript has a merit to be published in Journal of Fungi. To make this manuscript even better, please consider the following comments.
Authors: The authors are grateful for the reviewer’s kind comments and suggestions.
Reviewer 3: The authors need to mention in more detail whether a review article similar to the content of this manuscript was previously published. That will be useful information for the readers of this journal.
Authors: As far as the authors are aware, there are no others reviews with the same set of content as present herein. The published reviews about Scedosporium species are focused on pathogenesis mechanisms, immunology, treatment options, epidemiology, and taxonomy. In this way, a sentence explaining the difference between this review and the others previously published was added at the end of introduction.
Reviewer 3: The relevance of the topics covered in this review to the work of the authors to date should be mentioned in the introduction section.
Authors: The topics covered in the review were more detailed in the end of introduction.
Reviewer 3: The size of Figure 3 is too large. Also, for Non-Peptide Small-Molecule Metabolites and Non-Ribosomal Peptides in Figure 3, structural formulas of those compounds should be added.
Authors: The figure 3 is large because it contains many different classes of molecules released by Scedosporium. On the other hand, it is unviable to add the chemical structures of all molecules cited in Figure 3, because there are more than 40 molecules listed. But the structure of some molecules can be found in Figure 12 as examples. I hope the reviewer understands our positions on these regards.
Reviewer 3: The authors should add the structural formulas of the compounds listed in Table 1. That would be more beneficial to the readers of this journal.
Authors: The authors agree with the reviewer that it would be beneficial to the readers the presence of structure formulas of molecules; however, it is unviable to add all structural formulas expressed in the Table 1, since there are more than 100 molecules listed. For this reason, the structural formula of some of the most relevant molecules can be found in Figure 12 as examples. I hope the reviewer understands our positions in this regard.
Reviewer 3: The size of Figure 12 is too large. The authors should correct this.
Authors: The Figure 12 was resized.
Reviewer 3: If possible, the authors should be discussed the biosynthetic pathways of secondary metabolites and their relevance to pathogenicity.
Authors: Until this date we do not know much about the biosynthetic pathways of secondary metabolites and their relevance to pathogenicity on Scedosporium species. In fact, only a proposed biosynthetic pathway of acetylaranotin and boydin derivatives has been published (Le Govic et al., 2019 doi: 10.3389/fmicb.2019.02062).
Round 2
Reviewer 1 Report
The authors have presented satisfactory responses to my comments. One comment. I do not see the significance of including Figure 5. My interpretation is that data shown include larger proteins than in the previous version of the manuscript. That is fine, but there is no take home message except that the technique works in revealing lots of protein. This is preliminary data. Instead, the authors can state this as text.
Author Response
The authors have presented satisfactory responses to my comments. One comment. I do not see the significance of including Figure 5. My interpretation is that data shown include larger proteins than in the previous version of the manuscript. That is fine, but there is no take home message except that the technique works in revealing lots of protein. This is preliminary data. Instead, the authors can state this as text.
Authors: Firstly, the authors are grateful for the reviewer’s kind comments. The figure 5 has been included on the manuscript in order to illustrate the richness of proteins that can be found on the extracellular material secreted by Scedosporium; a phrase about this has been included in page 8. The data present in this figure 5 is the same as the previously demonstrated; however, as the genome of S. apiospermum has now been sequenced the number of identified proteins has increased.
Reviewer 2 Report
Although all my minor comments have been taken into account, I maintain my general comment. Regarding the list of secondary metabolites, a biological activity which potentially may improve the human or animal health has been reported for only a few compounds, and it would be more appropriate to focus on these compounds for this special issue. In addition, a large part of the manuscript deals with the physiology of these fungi without any real prospects of improving human or animal health. The fungal components or processes reviewed here are involved in the host-pathogen interplay, but in almost all cases, they have not been proven to be true virulence factors and therefore targets for the development of new antifungal drugs. To my knowledge, this is only the case for iron uptake since disruption of the gene encoding the non ribosomal peptide synthase involved in synthesis of the extracellular siderophore resulted in total loss of virulence. Finally, having a free waiver to be published is not an argument to explain the inadequacy of this review with the topic of the present special issue.
Author Response
Although all my minor comments have been taken into account, I maintain my general comment. Regarding the list of secondary metabolites, a biological activity which potentially may improve the human or animal health has been reported for only a few compounds, and it would be more appropriate to focus on these compounds for this special issue. In addition, a large part of the manuscript deals with the physiology of these fungi without any real prospects of improving human or animal health. The fungal components or processes reviewed here are involved in the host-pathogen interplay, but in almost all cases, they have not been proven to be true virulence factors and therefore targets for the development of new antifungal drugs. To my knowledge, this is only the case for iron uptake since disruption of the gene encoding the non ribosomal peptide synthase involved in synthesis of the extracellular siderophore resulted in total loss of virulence. Finally, having a free waiver to be published is not an argument to explain the inadequacy of this review with the topic of the present special issue.
Authors: The authors would like to cordially disagree with the reviewer appointment and we pointed out our position on this regard in the first round of revision. In this sense, we told the reviewer "The authors respectfully disagree with the reviewer, as the authors think that the topics addressed in the review are yes about the improvement of human life and health. This can be reflected in the topics about the virulence factors, which the studies can improve the clinical management of patients, as also the use of molecules secreted by the fungi for a better diagnosis. In addition, the use of fungal metabolites in degradation of pollutants as already appointed by the reviewer. In addition, the editor of the special issue previously approved the abstract that contained, when submitted for appreciation, all the points focused on our present review. So, the authors generously ask the reviewer to change its position regarding all these raised critical points. Also, the authors wish the reviewer to understand our comments and appeal."